# GALA: Geometry-Aware Local Adaptive Grids for Detailed 3D Generation

**Dingdong Yang**[1]   **Yizhi Wang**[1]   **Konrad Schindler**[2]   **Ali Mahdavi-Amiri**[1]   **Hao Zhang**[1]
[1]Simon Fraser University.   [2]ETH Zurich.

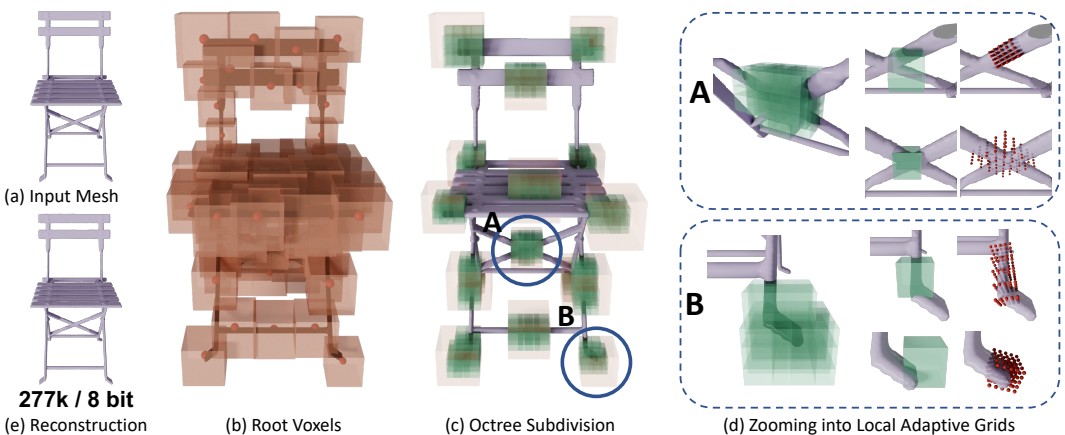

(a) Input Mesh

277k / 8 bit

(e) Reconstruction          (b) Root Voxels          (c) Octree Subdivision          (d) Zooming into Local Adaptive Grids

Figure 1: Given a watertight mesh (a), our representation, GALA, for *geometry-aware local adaptive grids*, distributes a set of root node voxels (coral) to cover the mesh *surfaces*. An octree subdivision is applied to each root, with a subset shown in (c). In each *non-empty* octree leaf node (green), a local grid (red dots) is oriented and anisotropically scaled to adapt to and tightly bound the local surface geometries. Only 277K parameters with 8-bit quantization yields an accurate representation (e).

## Abstract

We propose GALA, a novel representation of 3D shapes that (i) excels at capturing and reproducing complex geometry and surface details, (ii) is computationally efficient, and (iii) lends itself to 3D generative modelling with modern, diffusion-based schemes. The key idea of GALA is to exploit both the global sparsity of surfaces within a 3D volume and their local surface properties. *Sparsity* is promoted by covering only the 3D object boundaries, not empty space, with an ensemble of tree root voxels. Each voxel contains an octree to further limit storage and compute to regions that contain surfaces. *Adaptivity* is achieved by fitting one local and geometry-aware coordinate frame in each non-empty leaf node. Adjusting the orientation of the local grid, as well as the anisotropic scales of its axes, to the local surface shape greatly increases the amount of detail that can be stored in a given amount of memory, which in turn allows for quantization without loss of quality. With our optimized C++/CUDA implementation, GALA can be fitted to an object in less than 10 seconds. Moreover, the representation can efficiently be flattened and manipulated with transformer networks. We provide a cascaded generation pipeline capable of generating 3D shapes with great geometric detail. For more information, please visit our project page.

## 1 Introduction

The generation of high-quality 3D assets remains a costly task in terms of processing time and resources involving human efforts or machine compute. A continuing research pursuit is to develop efficient 3D representations to streamline the traditionally expensive 3D creation workflow

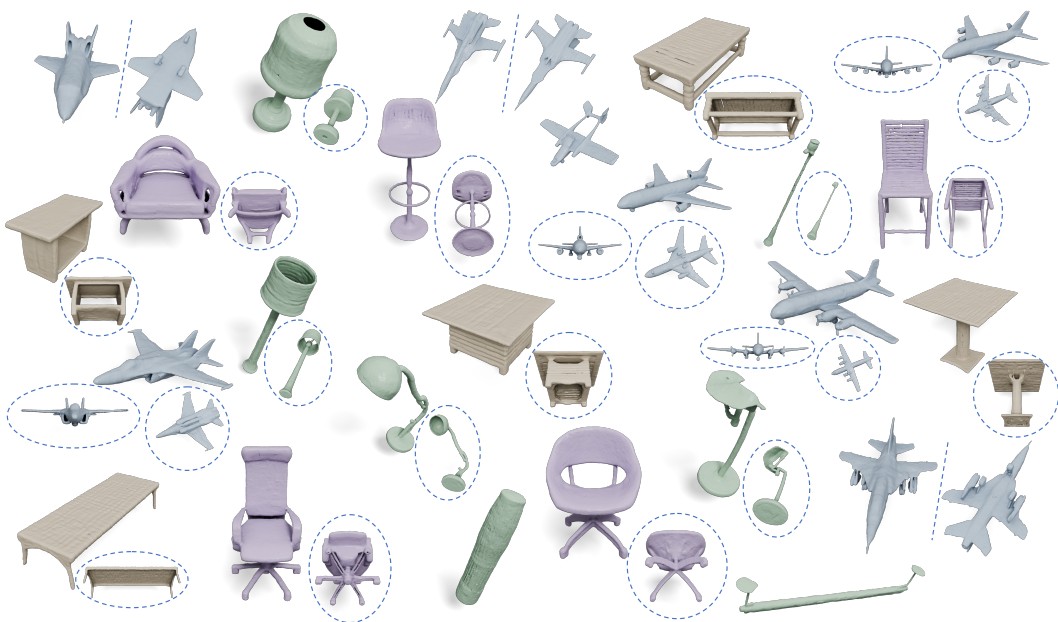

Figure 2: GALA enables diverse and detailed conditional 3D shape generation, including **Airplanes**, **Lamps**, **Tables** and **Chairs**. Best viewed on screen with high magnification.

Gupta et al. (2023); Jun & Nichol (2023); Li et al. (2023); Nichol et al. (2022); Shue et al. (2023), with the ultimate goal of speeding up the generative process without compromising the diversity and quality of the generated assets as reflected by their structural and geometric details.

To date, a variety of 3D representations including voxel (Li et al., 2023) and tetrahedral grids (Kalischek et al., 2024), meshes (Liu et al., 2023b), point clouds (Luo & Hu, 2021; Zhou et al., 2021), neural fields (Jun & Nichol, 2023), and triplanes (Chan et al., 2022; Shue et al., 2023; Gupta et al., 2023), have been employed for 3D generation, often leveraging powerful diffusion-based generators. However, as argued recently by (Yariv et al., 2023), none of these representations fulfill all of the three critical criteria for high-quality and scalable 3D generation: parameter efficiency for detail representation, preprocess efficiency for handling large datasets and mesh conversion, as well as simplicity of a tensorial form to suit modern-day neural processing. In response to this, they proposed Mosaic-SDF which encodes a signed distance function using a "mosaic" of aligned volumetric grids distributed over the shape surface, with different centers and different (still isotropic) scales.

In this paper, we accentuate that the key to 3D representation efficiency when using such an ensemble-of-tiles approach is *adaptivity* of the local coordinate grids to the geometry of the presented shapes. To this end, we propose to represent a 3D shape using a set of *geometry-aware* and *local adaptive* grids. Our novel representation, coined GALA, pushes the limit of adaptivity on three fronts:

- To exploit the global sparsity of typical object surfaces within a 3D volume, we place a *forest of octrees*, rather than uniform subdivisions, at the object boundaries, where a local grid is associated with each *non-empty* octree leaf node to store SDF values.
- Our local grids are not globally aligned. Instead, each grid may be differently oriented based on local PCA analysis of surface normals within the corresponding octree subdivision.
- Last but not least, we *anisotropically* scale each grid to better capture geometric details.

Our GALA representation can be implemented to yield sets of vectors that can be easily processed by transformer-based neural networks (Vaswani et al., 2017), while the octree forest provides a hierarchical 3D representation with limited depths. In Table 1, we compare GALA with several state-of-the-art representative methods over several useful criteria.

The contributions of our work are threefold:

1. Our geometry-aware, locally adaptive, and anisotropic grids enable more efficient (Table 1) and accurate sampling of shape structures, capturing geometric details, such as thin strings

and plates, which may be missed by both regular grid-based methods (Hui et al., 2022; Li et al., 2023) and Mosaic-SDF, whose grid tiles are neither shape- nor detail-aware.

2. We have developed an efficient implementation in pure CUDA and libtorch[1] for the entire GALA extraction process. This process costs about 10 seconds for each watertight input mesh on Nvidia A100 GPU, which is among the fastest ones of popular fitting methods, enabling fast data processing for large datasets.

3. The hierarchical structure of the octree forest in GALA facilitates the development of a practical cascaded 3D generation scheme with improved efficiency and quality.

| Methods | Description of representation | Parameter Count ↓ | Fitting Time ↓ | Representation Precision | Generation Network |
|---|---|---|---|---|---|
| S2VS Zhang et al. (2023) | Autoencoded vector set | ≤ 262k | - | Meidum/Low | Transformer |
| MeshDiffusion Liu et al. (2023a) | Deformable tetrahedral grid | ≥ 1.05M | 20m-30m | High/Medium | 3D UNet |
| Neural Wavelet Hui et al. (2022) | Two levels of wavelet coefficients | 536k | 1s | Medium | 3D UNet |
| Mosaic-SDF Yariv et al. (2023) | Grids on boundary | 355k | 2m | High | Transformer |
| GALA (ours) | Trees with adaptive grids on boundary | 277k | 10s | High | Transformer |

Table 1: Properties of different representations for 3D generation. The representation precision is defined as the reconstruction errors with respect to input meshes. See subsection 4.3 for more.

## 2 RELATED WORK

### 2.1 3D REPRESENTATION FOR GENERATION

We will first review the related work on 3D representation for generation, focusing on various methods characterized by shared features.

**Grid-based** Regular grid-based representations (Hui et al., 2022; 2024; Li et al., 2023) utilize a fixed-resolution grid for encoding various 3D quantities such as occupancy, signed distance function (SDF), color and etc. Such methods, however, may not capture extremely fine structures at the very initial stage of discretization. Additionally, regular grid-based representations allocate a substantial portion of their capacity to the empty spaces, leading to a potential cubic rate of parameter increase. In contrast, the adaptive local grids in GALA are able to capture thin structures and the octree forest allows for scaling the representation power with only a linear increase of parameters. Deep Marching Tetrahedra (Shen et al., 2021) (DMTet) uses deformable tetrahedral grids to improve the representation efficiency compared with regular grids. However, the fitting time (Liu et al., 2023a) of DMTet is about 10 times ours, the parameter count of DMTet is 3 times ours (See Table 1 for specific numbers.) while we represent geometry more accurately than it.

**Triplane** To mitigate the cubic complexity inherent in regular grid-based methods, triplane approaches (Chan et al., 2022; Shue et al., 2023; Gupta et al., 2023) employ three planes, each aligned with an axis in 3D space. Information at each 3D location is queried by first tri-linear interpolation from the three planes, followed by processing through various neural network architectures. Although triplane-based representations enhance parameter efficiency compared to regular grid-based methods, the tri-linear interpolation may result in oversmoothed outcomes and also missing geometric details.

**Point Cloud** Point cloud-based representations (Luo & Hu, 2021; Zhou et al., 2021; Zeng et al., 2022) describe 3D shapes by placing points on geometry boundaries. To accurately describe the shape, excessive number of points will be needed, which is not practical for the following generation task. Limited number of points will loss details of the shape. Moreover, post-processing methods (Kazhdan et al., 2006; Peng et al., 2021) are required to convert point clouds to meshes, which may introduce additional inductive bias.

**Neural Implicit** Neural implicit representations (Park et al., 2019; Chen & Zhang, 2019) model 3D shapes by learning mappings between coordinates and desired 3D quantities. When combined with the autoencoding framework, latent codes (Zhang et al., 2023; Chou et al., 2023) are derived. A drawback of these methods is that the autoencoding may result in oversmoothed geometric features. Furthermore, mesh extraction from neural implicit representations requires querying numerous points

---

[1]https://pytorch.org/cppdocs/

in space, yielding expensive neural network evaluations. In contrast, GALA enables direct and efficient extraction of geometries without the involvement of any neural network.

Lastly, we would address the work of Mosaic-SDF (Yariv et al., 2023) individually. Different from our method, Mosaic-SDF only has a single level of voxels and each voxel has a isotropic scale for each axis. In this sense, it has limited adaptation in representing the geometry. Moreover, it has to model the whole distribution as a whole at once while previous researches (Saharia et al., 2022; Pernias et al., 2023) have shown hierarchical or cascaded generation may benefit the generation.

## 2.2 Graph and Tree Generation

General graph generation relies on the Graph Neural Network and message passing mechanism. Others like HGGT (Jang et al., 2023) use graph compression methods such as $K^2-$Tree and sequentially generate tree nodes in an autoregressive way. In our case, generating an arbitrary topology graph is unnecessary because we focus on generating octrees. OGN (Tatarchenko et al., 2017) uses recursive way of generating octree voxels and Octree Transformer (Ibing et al., 2023) uses autoregressive way of generating tree nodes by transformers. In contrast with their methods, we use the widely adopted and effective generation paradigm, the diffusion model. BrepGen (Xu et al., 2024) generates fixed topology face-edge-vertice trees hierarchically by diffusion. Since the topology is fixed, the corresponding relation can be easily injected when training the diffusion model. We plan to adopt similar cascaded generation scheme to BrepGen based on our GALA representation.

## 3 Method

We will introduce our methods in two parts: subsection 3.1 the construction of GALA from the input watertight meshes and subsection 3.2 the cascaded generation of GALA.

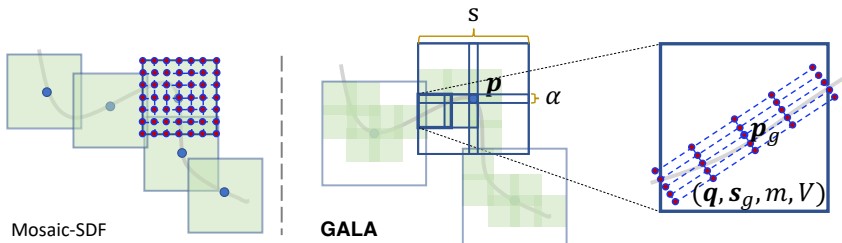

Figure 3: In our representation, **GALA**, tree root nodes, as voxels, are initialized over mesh surfaces (gray line), each with location $\mathbf{p} \in \mathbb{R}^3$ and scale $s \in \mathbb{R}$. Descendant node voxels are deduced recursively in the octree way, with overlapping, and expanded at ratio $\alpha \in \mathbb{R}$ into depth $d$. Only at a *non-empty* leaf node subdivision (light green) would a local adaptive grid of resolution $m \in \mathbb{N}^+$ be extracted with location $\mathbf{p}_g \in \mathbb{R}^3$, orientation quaternion $\mathbf{q} \in \mathbb{R}^4$ , scales $\mathbf{s_g} \in \mathbb{R}^3$, and values $V \in \mathbb{R}^{m^3}$. $N_o, \alpha, m, d$ are hyperparameters. In contrast, Mosaic-SDF (Yariv et al., 2023) (left) employs mosaic patches, each *fully* occupied by a *single-level*, *axis-aligned*, and *isotropic* grid.

## 3.1 Construction of GALA

We illustrate the overall GALA structure in 2D in Figure 3 and will dissect the construction process into: subsubsection 3.1.1 Octree forest initialization, subsubsection 3.1.2 Local adaptive grid extraction, subsubsection 3.1.3 Refinement and subsubsection 3.1.4 Fitting-aware Quantization.

### 3.1.1 Octree Forest Initialization

Given the number of trees $N_o$, we find the centroids $\mathbf{p}$ of tree root node voxels by Farthest Point Sampling (FPS) (Eldar et al., 1997) of densely sampled points $\mathcal{X}$ on the mesh of the input object. Then, to determine the scale $s$ of each root voxel $i$, we assign each sampled point $\mathbf{x} \in \mathcal{X}$ to its nearest centroid and define the scale scalar as the infinity-norm radius of each cluster:

$$s_i = \max_{\mathbf{x} \in \mathcal{C}_i} \|\mathbf{x} - \mathbf{p}_i\|_\infty, \quad \mathcal{C}_i = \left\{ \mathbf{x} \in \mathcal{X} \,\middle|\, i = \argmin_{k \in \{0,\dots,N_o-1\}} \|\mathbf{x} - \mathbf{p}_k\|_2 \right\} \tag{1}$$

In this way, we ensure that the forest will fully cover the shape boundaries. After setting the tree root voxels, we recursively deduce the further levels of octree nodes in an overlapping way where we slightly expand the divided spaces at each level by ratio $\alpha$. This expansion operation enables more voxels to contribute to the geometry modeling at early tree levels. See B.4 for the ablation study of $\alpha$.

### 3.1.2 LOCAL ADAPTIVE GRID EXTRACTION

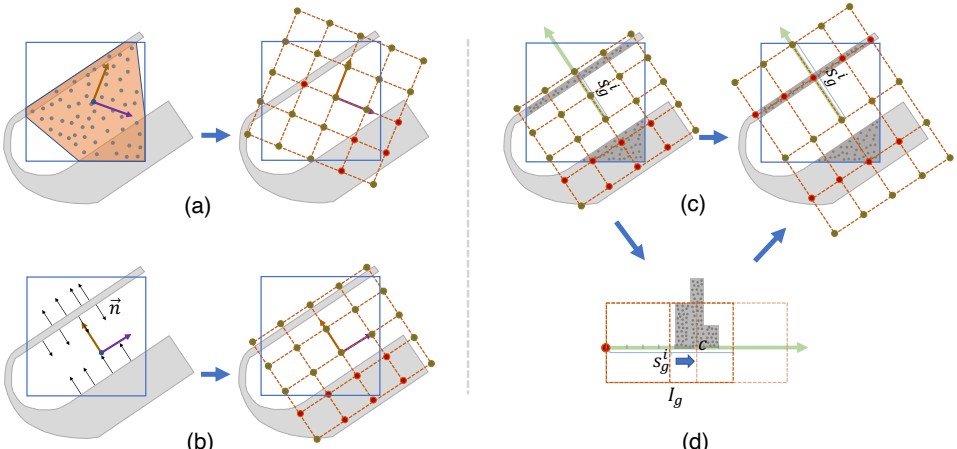

Figure 4: Illustration of local adaptive grid extraction in 2D. Within each subdivision (**blue square**), **(a)** an OBBTree (Gottschalk et al., 1996) determines the orientation of the local bounding box according to the **convex hull** of the subdivided **local geometry**. **(b)** Different from OBBTree, we determine the orientation of the **local adaptive grid** using PCA on bounded normal vectors ($\overrightarrow{n}$). **(c)** Moreover, we rescale the grid anisotropically to better capture the local geometry, as informed by the histogram **(d)** of sampled points (**small dark gray dots**) on triangle meshes projected along each **axis** of the grid. The •/• are grid samples with negative/positive signs.

Properly placing the local adaptive grid within each subdivision is essential to our method. The OBBTree (Gottschalk et al., 1996) determines the orientation of the local bounding box using eigenvectors of the covariance matrix derived from points sampled on the convex hull faces of the subdivided geometry (Figure 4(a)). However, we take a different approach. Instead of only defining a bounding box, we use a **grid** designed to capture as much geometric information as possible. Because the orientation derived from convex hull sample points may not align with the actual geometric structure, we align the grid orientation with PCA of the local normal vectors $\overrightarrow{n}$ instead (Figure 4(b)).

Additionally, we rescale the grid so that at least one set of grid samples locates at the most geometrically dense spot and adjacent grid samples with different signs capture more zero isosurfaces (Figure 4(c)). To achieve this, we utilize the histogram $\mathcal{H}$ of sampled points projected along each grid axis $\mathbf{u}_i$ and move nearest and smaller grid samples to the histogram peak (Figure 4(d), Algorithm 1)

---

**Algorithm 1** Rescaling the Grid with Histogram

1: Input: orientation $\mathbf{u}_i$ and scale $s_g^i$ at axis $i$; center $\mathbf{p}_g$ and bounded meshes $\mathcal{M}$
2: Split half axis of length $s_g^i$ into $n_h$ bins
3: Sample points $\mathbf{x} \in \mathcal{M}$
4: Count $\left| \frac{(\mathbf{x}-\mathbf{p}_g)\cdot\mathbf{u}_i}{\|(\mathbf{x}-\mathbf{p}_g)\|} \right|$ into histogram $\mathcal{H}$
5: $c \leftarrow$ center of $\max(\mathcal{H})$
6: Get grid index $I_g \leftarrow \argmax_I \left( \frac{2I}{m} < c \right)$
7: **if** $I_g > 0$ **then**
8: $\quad s_g^i \leftarrow \frac{m}{2I_g \cdot c}$
9: **end if**

---

We will show the effectiveness of our local adaptive grid extraction by ablation studies in section 4.5. After determining the locations, orientations and scales of anisotropic local grids, truncated ground truth SDF (Hui et al., 2022) values (truncated to $[-0.1, 0.1]$) are used to initialize the grid values

$V$. All the above initialization and extraction processes before refinement cost few seconds (as $2.56 \pm 1.30$s in ShapeNet Airplane). More detailed information on the time measurement, see A.4.

### 3.1.3 REFINEMENT

After extracting the local adaptive grids, we further refine the grid values $V$ by backpropagation to represent the input geometry more accurately. For each queried location $\mathbf{x}$, the queried value $v_{\mathbf{x}}$:

$$v_{\mathbf{x}} = \frac{1}{\sum_i w(\mathbf{x}, i)} \sum_i w(\mathbf{x}, i) \mathbf{I}(\mathbf{x}, i), \tag{2}$$

$$w(\mathbf{x}, i) = \text{ReLU}\left[1 - \|(\mathbf{x} - \mathbf{p}_g^i)\mathbf{O}_i \cdot \mathbf{s}_i^{-1}\|_\infty\right], \tag{3}$$

where $\mathbf{I}(\mathbf{x}, i)$ (Equation 2) is the tri-linear interpolation of grid values $V$ within the anisotropic local grid $i$ that encapsulates the query $\mathbf{x}$ and $w(\mathbf{x}, i)$ (Equation 3) is the weighting function which is defined by the $\ell_\infty$-norm neighborhood centered at the grid center $\mathbf{p}_g^i$, and oriented $\mathbf{O}_i \in \mathbb{R}^{3\times3}$ and scaled $\mathbf{s}_i^{-1}$ as the adaptive grid. All grid values will be optimized by MSE loss $L_{\text{MSE}} = \mathbb{E}(\|v_{\mathbf{x}} - v_{\text{GT}}(\mathbf{x})\|_2)$ using backpropagation, where $v_{\text{GT}}(\mathbf{x})$ is the truncated ground truth SDF value at location $\mathbf{x}$ directly computed from input meshes. We assign $v_{\mathbf{x}} = 0.1$ if $\mathbf{x}$ does not lie in any local grid. We will flip signs of interior parts in slices $\mathbb{R}^{\ell\times\ell}$ of the SDF extraction grid $\mathbb{R}^{\ell\times\ell\times\ell}$ to $-0.1$ via a DFS method when extracting meshes from GALA. Mesh extraction algorithm is detailed in A.1.

### 3.1.4 FITTING-AWARE QUANTIZATION

To increase the storage (see more at B.3) and memory efficiency of GALA as well as lower the difficulty of aligning float values in space inspired by the discrezation step in MeshGPT (Siddiqui et al., 2024), we quantize the local adaptive grid related quantities $\mathbf{q}, \mathbf{p}_g, \mathbf{s_g}, V$ during the fitting process by Straight-Through Estimator (STE) (Bengio et al., 2013; Han et al., 2015). We quantize the orientation in the unit of $\frac{\pi}{60}$ of Euler angles and $\mathbf{s}_g, \mathbf{p}_g, V$ in 8 bit of the range $[0, 0.1], [-0.5, 0.5], [-0.1, 0.1]$.

## 3.2 CASCADED GENERATION OF GALA

Motivated by previous cascaded generation works (Saharia et al., 2022; Ho et al., 2022; Xu et al., 2024; Hui et al., 2022; Chen et al., 2021) which successfully generate complicated 2D or 3D data in a coarse-to-fine manner, we generate sets of information $X_o, X_{\bar{V}}, X_V$, of GALA step by step where flattened vectors of root node $X_o = \{\mathbf{x}_o \in \mathbb{R}^4 | (\mathbf{p}, s)\}$, local adaptive grid configuration $X_{\bar{V}} = \{\mathbf{x}_{\bar{V}} \in \mathbb{R}^{10} | (\mathbf{q}, \mathbf{s}_g, \mathbf{p}_g)\}$ and grid values $X_V = \{\mathbf{x}_V \in \mathbb{R}^{m^3}\}$ are composed (Refer to A.2 for more details on the data preparation). Specially, since the configurations of local adaptive grids $X_{\bar{V}}$ already encode rich local geometric information as articulated in subsubsection 3.1.2, we are able to directly regress $X_V$ from $X_{\bar{V}}$, which greatly alleviates the generation training burden.

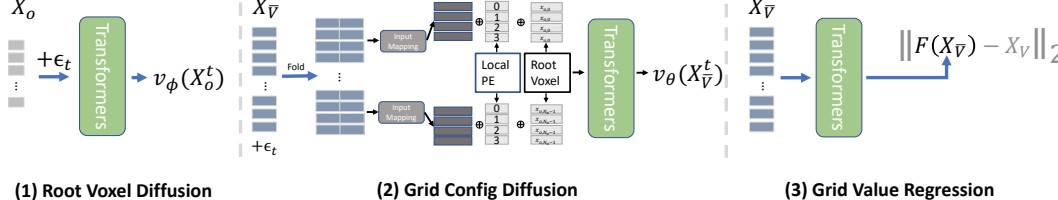

(1) Root Voxel Diffusion      (2) Grid Config Diffusion      (3) Grid Value Regression

Figure 5: GALA generates shapes in a cascaded manner by (1) Root voxel diffusion; (2) Local adaptive grid configuration diffusion; (3) Local adaptive grid value prediction by regression.

All sets of vectors are processed by transformer-based networks (Vaswani et al., 2017; Peebles & Xie, 2023). (1) The class label $y$ conditioned distribution of root voxels $P(X_o|y)$ is firstly modeled by diffusion model (Song et al., 2020; Ho et al., 2020). (2) Configurations of anisotropic local grids $X_{\bar{V}}$ are generated following that via modeling $P(X_{\bar{V}}|X_o, y)$ by diffusion model. (3) Lastly, grid values $V$ are predicated from grid configurations $\bar{V}$ via $X_V = F(X_{\bar{V}}, y)$ and trained by L2 regression loss. More specifically speaking, as we can see in Figure 5.(2), siblings of padded leaves are grouped and

folded once and previously generated root voxel information $x_{o,i} \in X_o$ is added directly to the nodes of its corresponding descendants together with local positional encodings (PE) of trees to differentiate siblings among them. In all, following the popular diffusion training paradigm (Ho et al., 2020; Song et al., 2020), we update the diffusion network weights $\theta$ via the v-prediction (Salimans & Ho, 2022):

$$\nabla_\theta \| v - v_\theta(X_V^t, t, y, X_o) \|^2, \quad v = \alpha_t X_V^0 - \sigma_t \epsilon \tag{4}$$

where $X_{\bar{V}}^t = \sqrt{\bar{\alpha}_t} X_{\bar{V}}^0 + \sqrt{1 - \bar{\alpha}_t}\epsilon, X_{\bar{V}}^0 \sim q(X_{\bar{V}}), \epsilon \sim \mathcal{N}(\mathbf{0}, \mathbf{I})$ is the diffusion forward process. $\bar{\alpha}_t$ is defined by $\bar{\alpha}_t = \prod^t \alpha_t, \alpha_t = 1 - \beta_t$ where $\beta_t$ is the variance schedule. $\sigma_t = \sqrt{1 - \bar{\alpha}_t}$ and cosine noise scheduling (Nichol & Dhariwal, 2021) is used. The network weights $\phi$ for the first step are updated in a similar manner as described above. For more details, please refer to A.5.

## 4 EXPERIMENTS AND RESULTS

### 4.1 IMPLEMENTATION DETAILS

For the hyperparameter of GALA, we set tree root number $N_o = 256$, child node expanding ratio $\alpha = 0.2$, grid resolution $m = 5$ and depth $d = 1$. The number of histogram bins $\mathcal{H}$ is $2m$. For the refinement process, 8192 points are queried near surfaces on each run for 400 iterations. Details of the pure C++/CUDA implementation and anonymous code release link will be found at A.3.

For the cascaded generation, each network for every generation step is consisted of 24 transformer layers with model channel 1024. AdamW (Loshchilov & Hutter, 2017) with lr $= 8e^{-5}, \beta_1 = 0.9, \beta_2 = 0.999$ is adopted. We test the reconstruction and generation ability of our methods on the whole ShapeNet (Chang et al., 2015) dataset. We follow the dataset split of previous works (Zhang et al., 2022; 2023; Yariv et al., 2023) and train conditional generation on the whole ShapeNet classes.

### 4.2 METRICS

We evaluate reconstruction results using Chamfer Distance (CD) between two sampled 3D objects and also report the Light Field Distance (LFD) (Chen et al., 2003), a widely adopted (Chen et al., 2020; Gao et al., 2022; Zuo et al., 2023) visual similarity metric that is well-known from computer graphics. To evaluate the generation results, we adopt the metric of Maximum Mean Discrepancy (MMD), Coverage (COV), and 1-nearest neighbor accuracy (1-NNA) based on Chamfer Distance (CD) and Earth Mover Distance (EMD) (Luo & Hu, 2021; Zeng et al., 2022).

### 4.3 RECONSTRUCTION RESULTS

We compare our reconstruction results with 3DShape2VecSet (S2VS) (Zhang et al., 2023), Neural Wavelet (NW) (Hui et al., 2022) and MeshDiffusion (DMTet, $64^3 \times 4$) (Liu et al., 2023a). We measure the reconstruction metrics on 50 objects of ShapeNet classes (airplane, car, char, basket, guitar), 10 objects per class. From Table 2, we can see that our reconstruction is more accurate than other baselines. $\mathcal{A}$ means local adaptive grids. Without $\mathcal{A}$, uniform grid samplings will be placed within tree voxels in the axis-aligned way. Visual comparisons on results w/o $\mathcal{A}$ are shown in B.1, where the local adaptive grid setting captures more accurate details on geometry surfaces. Qualitative comparison of reconstruction is shown in Figure 6. As we can see from Figure 6, we preserve the most geometric details with one of the fewest parameter numbers.

| Metric | S2VS | NW | MeshDiffusion | Ours, w/o $\mathcal{A}$ | Ours |
|---|---|---|---|---|---|
| LFD $\downarrow$ | 2429.92 | 1869.80 | 1301.14 | 1146.74 | 1109.36 |
| CD $\downarrow$ ($\times 10^{-3}$) | 4.32 | 4.17 | 2.61 | 1.52 | 1.18 |
| #Parameters | $\leq$ 262k | 536k | 1.05M | 263k | 277k |

Table 2: Reconstruction quality across 5 ShapeNet classes (airplane, basket, car, chair, guitar; 10 random objects per class).

Because we do not have the official code release from Mosaic-SDF (Yariv et al., 2023), we requested the reconstruction results from the authors of Mosaic-SDF and compare the reconstruction of ours

with them qualitatively in Figure 7. As we can see from Figure 7, the car is a challenging example shown in Mosaic-SDF and with complicated exterior and interior structures such as very thin and close surfaces. GALA can capture exterior and interior geometry structures at the same time. In contrast, the result of Mosaic-SDF fails to capture internal structures. As for the guitar example, GALA has the better connection of guitar strings and the thickness of strings is closer to the ground truth than it of Mosaic-SDF.

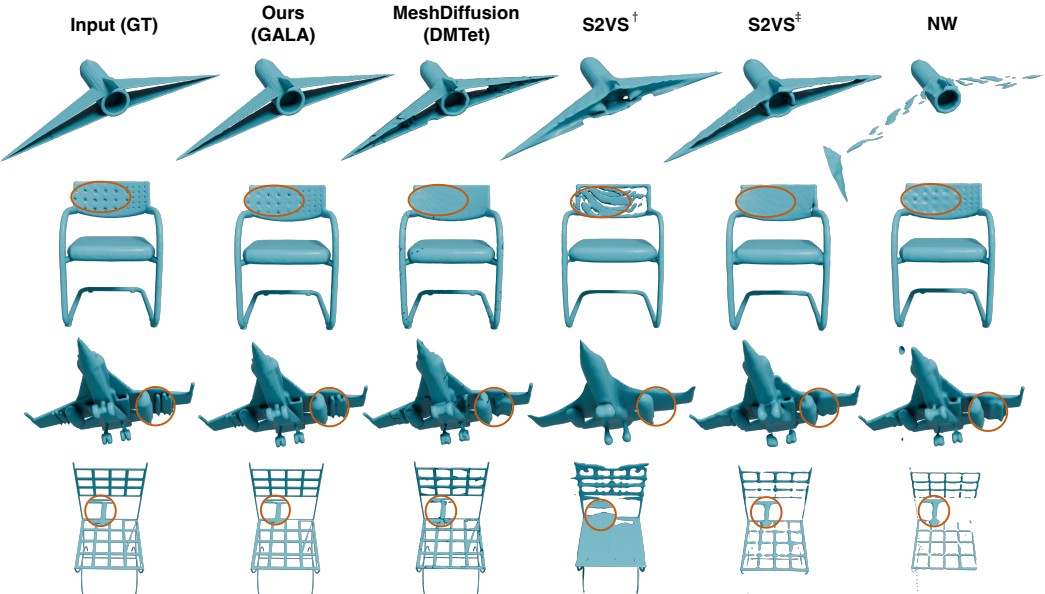

Figure 6: Qualitative comparison of reconstruction results between ours and other important baselines. S2VS[†] (Zhang et al., 2023) is evaluated from the public checkpoint provided in the official repository and S2VS[‡] is our own implementation. All evaluated at resolution $256^3$. Zoom in to see more details.

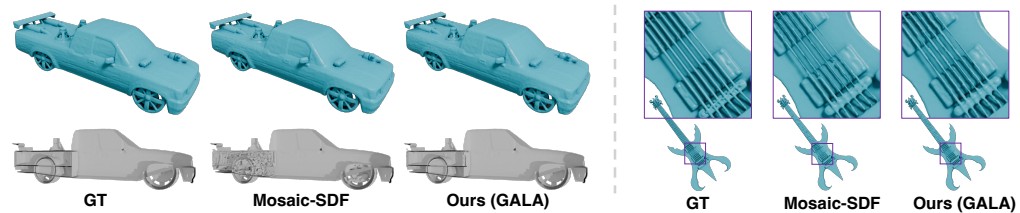

Figure 7: Qualitative comparison of the reconstruction result with Mosaic-SDF (Yariv et al., 2023). Mosaic-SDF results are provided by the author directly. We reveal the complicated internal structures of the input GT and the reconstructed results by slicing them based on Slice3D (Wang et al., 2023).

## 4.4 GENERATION RESULTS

| | airplane | | | | | | chair | | | | | |
| | COV(↑,%) | | MMD(↓) | | 1-NNA(↓,%) | | COV(↑,%) | | MMD(↓) | | 1-NNA(↓,%) | |
| | CD | EMD | CD | EMD | CD | EMD | CD | EMD | CD | EMD | CD | EMD |
|---|---|---|---|---|---|---|---|---|---|---|---|---|
| 3DIGL | 41.09 | 32.67 | 4.69 | 4.73 | 82.67 | 84.41 | 37.87 | 39.94 | 20.37 | 10.54 | 74.11 | 69.38 |
| NW | 51.98 | 45.05 | 3.36 | 4.19 | 68.32 | 73.76 | 43.79 | 47.04 | 16.53 | 9.47 | 59.47 | 64.20 |
| S2VS | 51.98 | 40.59 | 3.80 | 4.45 | 69.06 | 76.73 | 51.78 | 52.37 | 16.97 | 9.44 | 58.43 | 60.80 |
| Mosaic-SDF | 52.48 | 51.49 | 3.54 | 3.78 | 62.62 | 69.55 | 48.22 | 55.03 | 15.47 | 9.13 | 51.04 | 55.62 |
| Ours | 52.91 | 51.49 | 2.48 | 2.84 | 59.90 | 57.43 | 52.21 | 53.10 | 13.28 | 8.17 | 53.10 | 54.57 |

Table 3: Generation quality for ShapeNet classes of airplane and chair. By convention, MMD-CD has been multiplied by $10^3$ and MMD-EMD by $10^2$ for readability.

From Table 3, we show our cascaded generation on GALA produces shapes better than other baselines quantitatively. We also provide qualitative results on Figure 8 to show that our generation gives much geometry details in generation results. Please see the supplementary file for more generation results.

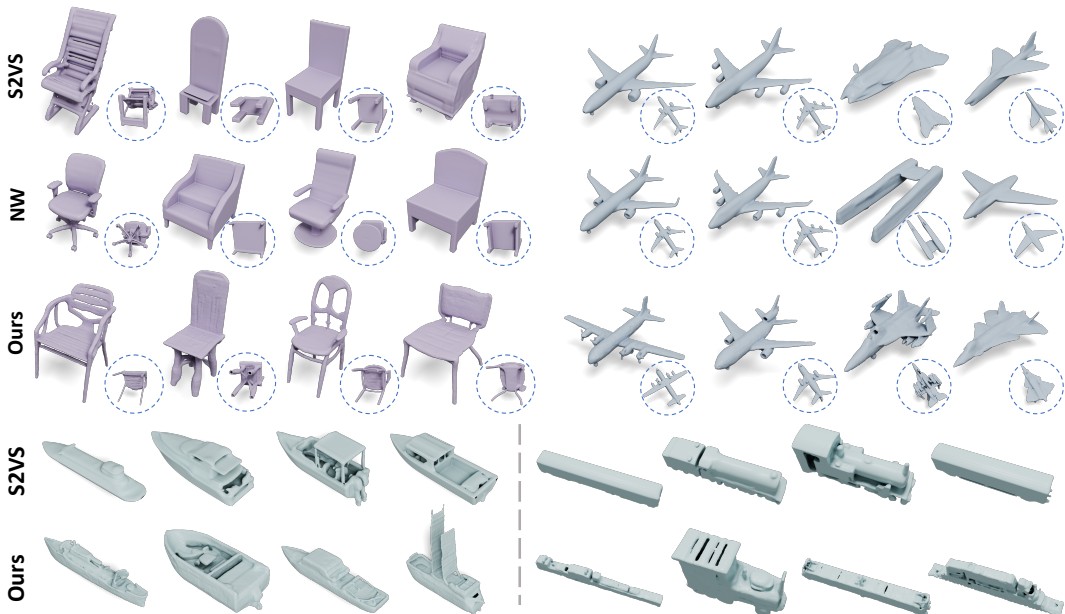

Figure 8: The qualitative comparison of generation results. Additional results of ShapeNet vessel and train are shown between S2VS and ours to demonstrate the detailed generation of ours.

## 4.5 ABLATION STUDIES

**Local Grid Extraction.** We conduct the ablation study on the components in local adaptive grid extraction, including determining orientation by bounded normals ($\overrightarrow{n}$) and rescaling with histogram ($\mathcal{H}$). Table 4 shows the effectiveness of our local adaptive grid extraction design and limited degradation by the quantization procedure. We further give some qualitative examples in Figure 9 to show that corresponding components help better capture the detailed geometries.

| Metrics | vanilla | $+ \overrightarrow{n}$ | $+ \overrightarrow{n}, \mathcal{H}$ |
|---|---|---|---|
| CD $\downarrow (\times 10^{-3})$ | 1.55 | 1.33 | 1.18 (1.08†) |
| HD $\downarrow (\times 10^{-2})$ | 1.46 | 1.19 | 1.06 (0.98†) |

Table 4: Ablating the anisotropic local grid extraction components ($\overrightarrow{n}, \mathcal{H}$) on the same set of data of reconstruction test. †: without quantization.

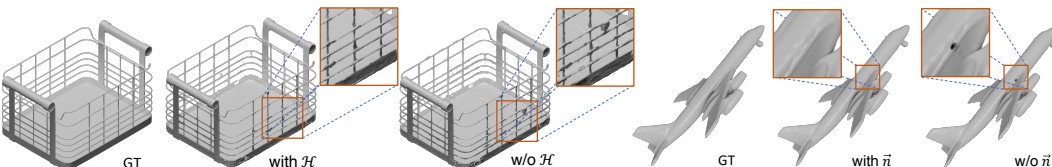

Figure 9: Qualitative examples of GALA fitting results on **Left**: rescaling the local grid with histogram or not; **Right**: determining the grid orientation with normals or not. Zoom in to see more.

**More Tree Setting.** We examine the effect of varying tree counts $N_o$ (with depth $d = 2$) on three examples, as shown in Figure 10. As we can see in Figure 10, using $N_o = 32$ with fewer than 200k parameters already yields low-error reconstructions across the three distinct example objects. Furthermore, even at $N_o = 16$ and fewer than 100k parameters, our method effectively captures significant geometric details, such as the missiles on the airplane's rear and the lamp's strings, demonstrating the method's great adaptability.

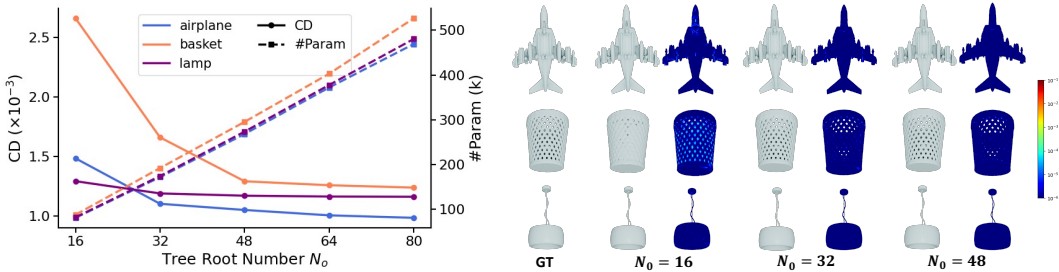

Figure 10: Influence of tree count $N_o$ (with depth $d = 2$) on three examples.

### 4.6 NOVELTY OF GENERATED RESULTS

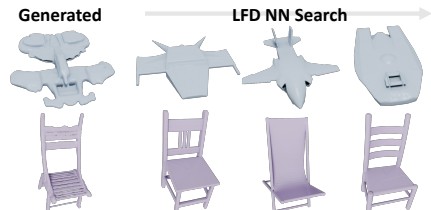

In Figure 11, we show two shapes generated by our method and three closest training shapes retrieved via nearest neighbour search (NN) using LFD (Chen et al., 2003). We can see that our generations have detailed and distinct geometric structures from those in the training set, demonstrating our method's generative novelty.

Figure 11: Two generated shapes and three visually closest (left to right) training shapes retrieved using LFD.

### 4.7 APPLICATION: AUTOCOMPLETION

Due to the hierarchical structure of GALA and the cascaded generation pipeline, we are able to roughly outline the objects to be generated using coarse and partial root voxels. Based on the work RePaint (Lugmayr et al., 2022), we autocomplete the given root voxels by substituting intermediate results at timesteps with the given ones during denoising process. See Figure 12 for the results.

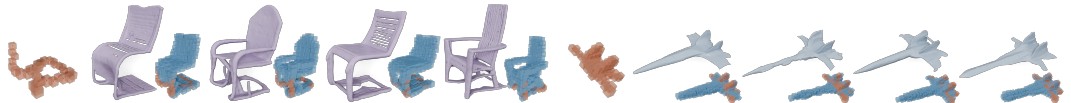

Figure 12: Given the partial root voxels (orange), we autocomplete the rest (blue) and generate geometries.

### 4.8 TEXTURING THE GENERATED MESH

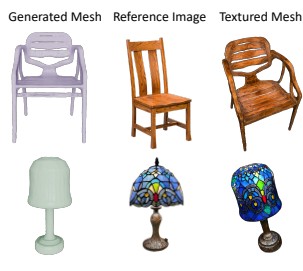

First carving the mesh and then adding textures is a standard workflow in 3D asset creation within the industry. It allows flexibility of creating various textures while reusing the fine meshes already crafted. Many previous works (Zhang et al., 2024; Perla et al., 2024; Cao et al., 2023; Chen et al., 2023; Richardson et al., 2023) in deep learning also adopt similar strategies. Therefore, to show our generated meshes can be easily textured, in Figure 13, we use Easi-Tex (Perla et al., 2024) to texture examples of our generated meshes by reference images.

Figure 13: Texturing generated meshes.

## 5 CONCLUSION AND FUTURE WORK

We present GALA, geometry-aware local adaptive grids for detailed 3D generation. We demonstrate that GALA can represent detailed geometries efficiently with fewer parameters. Additionally, we propose a cascaded generation approach for GALA that produces detailed geometry. We also deliver an efficient C++/CUDA implementation of GALA that completes fitting in seconds, significantly enhancing the practical usage of our method. For future work, compact graph representations (Bouritsas et al., 2021; Jo et al., 2022; Jang et al., 2023) may facilitate tree compression within the forest and further ease the generation step. Also of interests is to remove the watertight constraint on the input meshes and extend GALA and the ensuring generation to unsigned distance fields (UDFs).

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

# A  IMPLEMENTATION DETAILS

## A.1  FLIP SIGNS OF THE INTERIOR PARTS AND EXTRACT MESH

Figure 14: Flip signs of 0.1 *islands* (connected components) surrounded by negatives.

We will flip the signs of **connected components of 0.1 surrounded by negative values** in a SDF slice $S \in \mathbb{R}^{\ell \times \ell}$ (slicing along z-axis) extracted from GALA, where $\ell$ is the extracted SDF grid ($\mathbb{R}^{\ell \times \ell \times \ell}$) resolution. It is a Depth-first Search (DFS) algorithm as described in Algorithm 2. This post-processing procedure will eliminate redundant geometry blobs in interior parts of objects as illustrated in Figure 14. After getting all the slices, we can concatenate them together to form the grid and use Marching Cube (Lorensen & Cline, 1998) to extract mesh from it.

---

**Algorithm 2** Flip Signs of the Interior Parts

---

1: Input: A fitted GALA, $z$ and $\ell$
2: Extract SDF slice $S \in \mathbb{R}^{\ell \times \ell}$ at $z$ from GALA           ▷ According to Eq (2,3) of the main text.
3: Initialize visited matrix $V$ of size $\ell \times \ell$ to false
4: Define directions $D = \{(0, 1), (0, -1), (1, 0), (-1, 0)\}$
5: **for** $i = 0$ to $\ell - 1$ **do**
6:     **for** $j = 0$ to $\ell - 1$ **do**
7:         **if** $|S[i, j] - 0.1| <$ EPS  &  $\neg V[i, j]$ **then**     ▷ Check possible islands of 0.1 SDF values.
8:             $stack \leftarrow \emptyset$
9:             $stack$.push$((i, j))$
10:            $V[i, j] \leftarrow$ true
11:            $isSurrounded \leftarrow$ true
12:            $positions \leftarrow \emptyset$
13:            **while** not $stack$.empty() **do**
14:                $(x, y) \leftarrow stack$.pop()
15:                $positions$.add$((x, y))$
16:                **for all** $(dx, dy) \in D$ **do**
17:                    $(nx, ny) \leftarrow (x + dx, y + dy)$
18:                    **if** $nx < 0 \vee nx \geq \ell \vee ny < 0 \vee ny \geq \ell$ **then**           ▷ Boundary condition.
19:                        $isSurrounded \leftarrow$ false
20:                        **continue**
21:                    **end if**
22:                    **if** $|S[nx, ny] - 0.1| <$ EPS  &  $\neg V[nx, ny]$ **then**     ▷ Push 0.1 to the island.
23:                        $stack$.push$((nx, ny))$
24:                        $V[nx, ny] \leftarrow$ true
25:                    **end if**
26:                    **if** $S[nx, ny] \geq 0$  &  $|S[nx, ny] - 0.1| >$ EPS **then** ▷ Check other positives.
27:                        $isSurrounded \leftarrow$ false
28:                    **end if**
29:                **end for**
30:            **end while**
31:            **if** $isSurrounded$ **then**
32:                **for all** $(x, y) \in positions$ **do**
33:                    $S[x, y] \leftarrow -S[x, y]$                          ▷ Flip the sign of the 0.1 island.
34:                **end for**
35:            **end if**
36:        **end if**
37:    **end for**
38: **end for**

---

## A.2 DATA PREPROCESSING

For each root or non-empty leaf tree node, information will be saved as illustrated in Figure 15. The voxel position $\mathbf{p}$ and scale $s$ (Figure 15.**Right**) would only be recorded and generated at the root level. The $\mathbf{p}, s$ of subsequent levels of tree nodes will be deduced deterministically from the root voxels. The parent index and sibling index (Figure 15.**Left**) is only recorded for recover the tree structure and will not be explicitly generated. Quaternion is used to represent the orientation of the local grid. Moreover, local grid locations and scales are stored and generated. All the quantities within the tree node vector will be normalized to [0,1] by $\min, \max$ statistics calculated within each data domain $(\mathbf{p}, s, \mathbf{p}_g,\text{etc})$ of specific datasets.

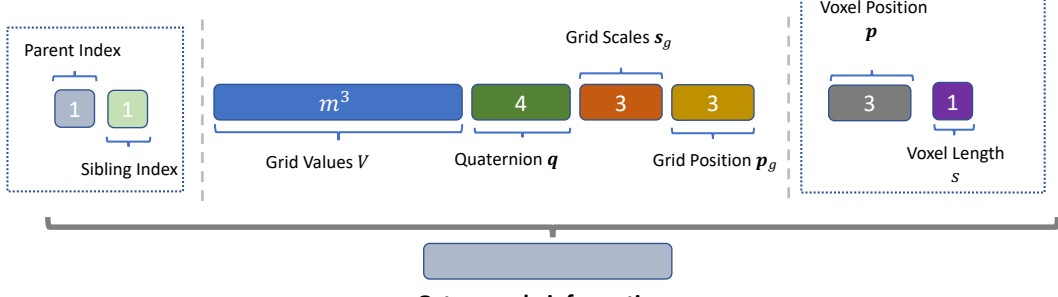

Figure 15: **Octree Node Information**. **Left**: the *Parent Index* $\in [0, \dots, N_0 \times 8^{l-1} - 1]$ and *Sibling Index* $\in [0, \dots, 7]$, $l$ is the tree level and $N_0$ is the tree root number. When $l = 0$, placeholder values are put on the two indices; **Middle**: local adaptive grid related information, the scales $\mathbf{s}_g$ and position $\mathbf{p}_g$ of the grid are stored; **Right**: the values of root voxels $\mathbf{p}, s$.

The GALA of each object will be flattened and stored as the format illustrated in Figure 16.

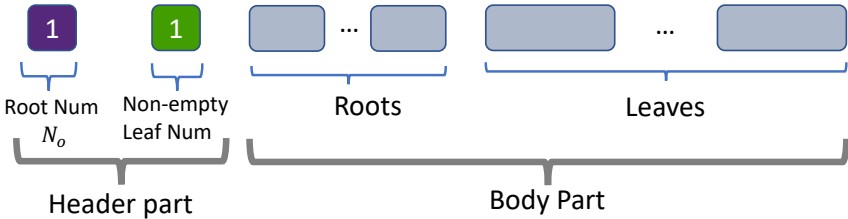

Figure 16: The data format of GALA to be flattened and stored.

## A.3 DETAILS OF THE PURE C++/CUDA IMPLEMENTATION

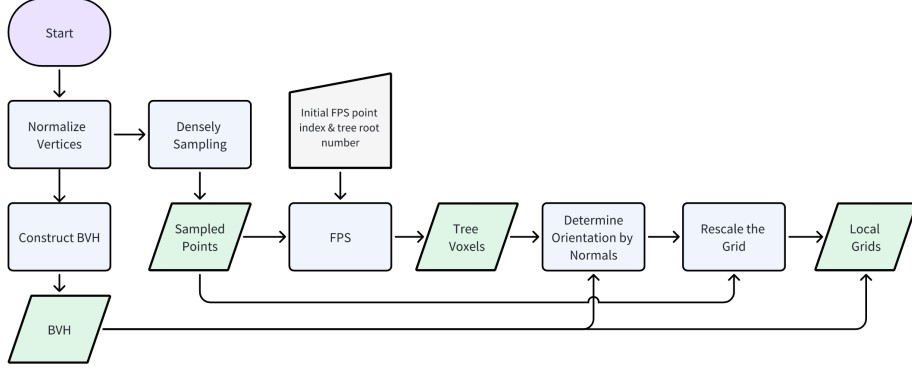

Figure 17: The algorithm flowchart of our GALA extraction.

We show the algorithm flowchart Figure 17 of the initialization stage of GALA, which is built upon CUDA, Libtorch, a third-party R-Tree based SDF query library[2] and few head only utility libraries such as C++ argparse[3]. We will dig into the details of the implementation step by step as follows.

**Normalize Vertices** We shift the vertices by the center location $(( \frac{x_{\max}+x_{\min}}{2}, \frac{y_{\max}+y_{\min}}{2}, \frac{z_{\max}+z_{\min}}{2} ))$ and then scale them by the diagonal distance ($\sqrt{(x_{\max} - x_{\min})^2 + (y_{\max} - y_{\min})^2 + (z_{\max} - z_{\min})^2}$).

**Construct Bounding Volume Hierarchy (BVH)** In order to quickly search relevant mesh information in the space, we construct Bounding Volume Hierarchy (BVH) on GPU parallelly via radix tree according to (Karras, 2012). We encountered some bug when doing the GPU parallel bottom-up AABB (Axis-Aligned Bounding Box) construction of the radix tree where very few random items of AABB ($\sim 0.01\%$) at z-dimension were randomly corrupted for each run. To circumvent this, we copy the whole tree back to CPU, conduct a DFS post-order traverse to construct the AABB and copy the results back to GPU.

**Densely Sampling Points On Meshes** We sample $10 \times N_t$ points on meshes where $N_t$ is the number of triangles of the input mesh. We randomly sample points on triangles using barycentric coordinates. The number of points sampled at each triangle is proportional to its surface area.

**Initial Point of Farthest Point Sampling** As described in the main text, in order to augment our representation, the extraction algorithm also accepts the index of the initial FPS point. This enables possible data augmentation of GALA representation by choosing different initial FPS points for the same input mesh.

**Determining Orientation by Normals** As detailed in the main text, we estimate the orientation of the local grid using the eigenvectors of the covariance matrix of the bounded normal vectors. We get the eigenvectors in each local grid in parallel on each GPU thread using Jacobi eigenvalue algorithm (Golub & Van Loan, 2013), where the maximum iteration is set as 50 and the tolerance for maximum off-diagonal value is $1e^{-10}$.

### A.4 TIME MEASUREMENT OF GALA FITTING

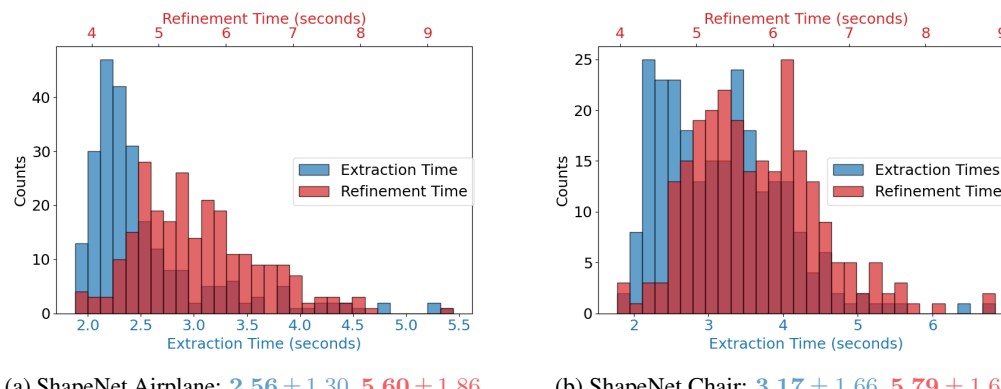

(a) ShapeNet Airplane: $\mathbf{2.56} \pm 1.30$, $\mathbf{5.60} \pm 1.86$  (b) ShapeNet Chair: $\mathbf{3.17} \pm 1.66$, $\mathbf{5.79} \pm 1.67$

Figure 18: The fitting time statistics of GALA on ShapeNet Airplane and Chair class.

In this section, we present detailed time measurements of our GALA fitting process. The measurements were conducted under the configuration of 6 virtualized logical cores (hyper-thread) of AMD EPYC 7413 @2.65GHz and 1 Nvidia A100. We show the fitting time statistics plots as in Figure 18 of 250 objects from ShapeNet Airplane and ShapeNet Chair respectively. The extraction (blue) time includes all initialization and local adaptive grid extraction process, namely all the procedures other than the refinement. Based on the comprehensive statistics shown in Figure 18, we show that our implementation indeed achieves very fast fitting in less than 10 seconds overall, as stated in the main text. Additionally, the peak GPU memory usage during GALA fitting is recorded at less than 800MB.

---

[2]https://github.com/sxyu/sdf
[3]https://github.com/p-ranav/argparse

Since the SDF query algorithm supports multi-threading, more CPU logical cores will further decrease the fitting time. In a 30 logical cores (virtualized) AMD EPYC 7J13 and 1 Nvidia A100 machine, the total fitting time will be less than 6 seconds. We also provide a reference record on a lower-end local desktop machine, configured with 1 Nvidia RTX 3060 and Intel i9-12900K, showing approximately 3 seconds for extraction and 27 seconds for refinement.

## A.5 MORE DETAILS ON CASCADED GENERATION

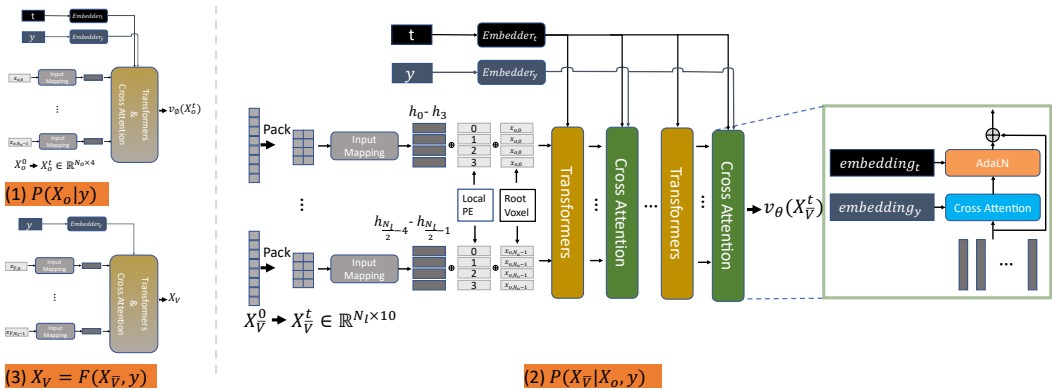

Figure 19: The network details of the cascaded generation pipeline, composed by steps: (1) root voxel diffusion $P(X_o|y)$ by diffusion; (2) Local adaptive grid configuration generation $P(X_{\bar{V}}|X_o, y)$ by diffusion; (3) Local adaptive grid value prediction $X_V = F(X_{\bar{V}}, y)$ trained by regression.

As shown in Figure 19, our generation networks are based on transformer networks (Vaswani et al., 2017). The root voxel conditions and local positional encodings of the tree are injected in an in-context manner through addition, while the timestep condition is injected using the adaLN-Zero block from DiT (Peebles & Xie, 2023), and the object class condition is injected via cross-attention. Additionally, noise augmentation is applied, as in previous works (Ho et al., 2022; Xu et al., 2024; Saharia et al., 2022), to mitigate the impact of the training/inference discrepancy in cascaded generation. Furthermore, zero-terminal (Lin et al., 2024) is adopted to ensure the last timestep of diffusion starts from a zero signal-to-noise ratio (SNR). Classifier-free guidance (Ho & Salimans, 2022) is used, with an additional null label trained as $y$ during diffusion training, and $w = 0$ is applied during classifier-free guidance in diffusion inference.

# B OTHER ABLATION STUDIES

## B.1 MORE ILLUSTRATION ON W/O $\mathcal{A}$

As we can see from Figure 20, since the anisotropic local grid setting ($\mathcal{A}$) enables close and adaptive sampling on object surfaces, the full setting of GALA captures much more accurate and artifact-free surfaces than w/o $\mathcal{A}$.

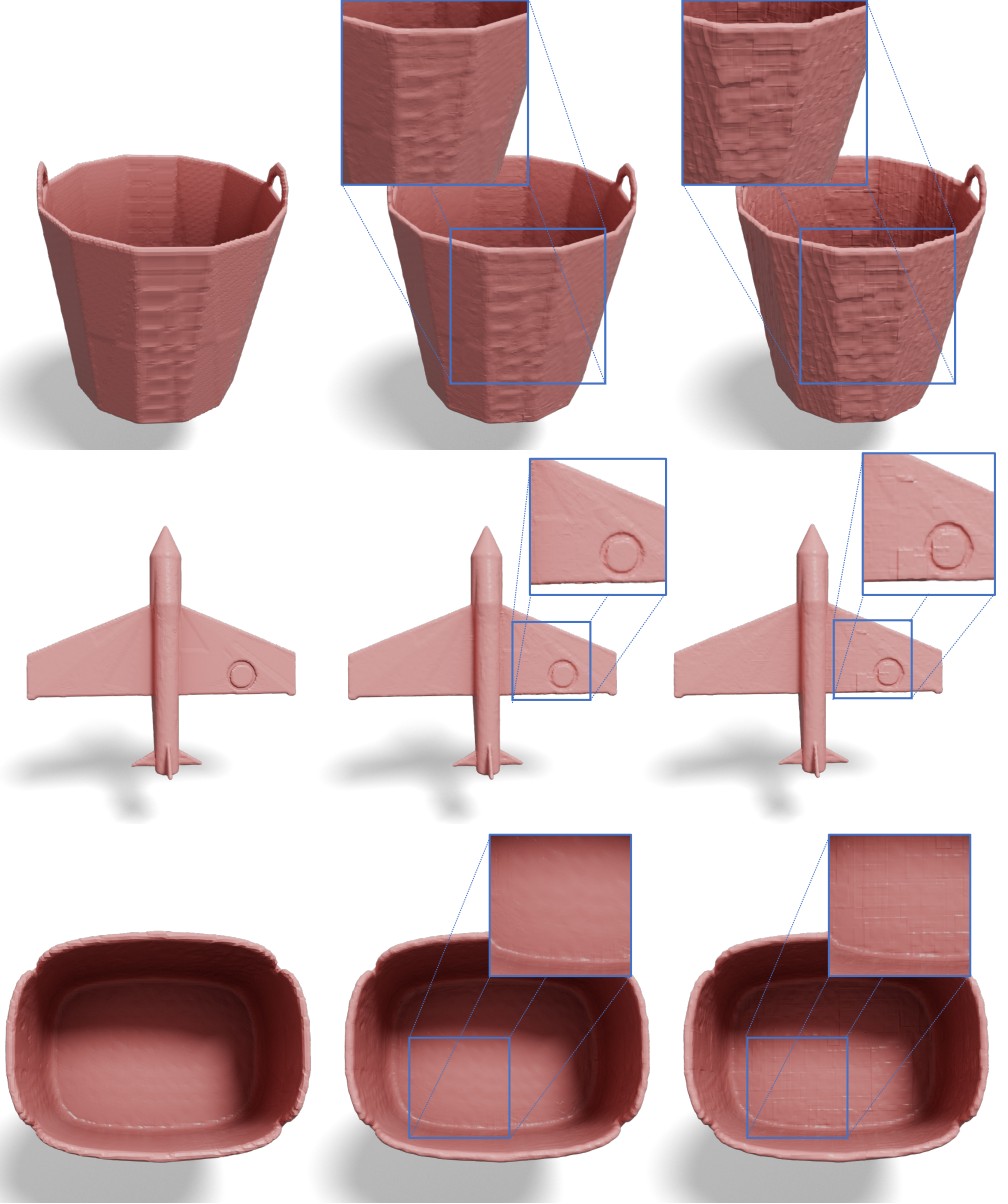

Figure 20: Qualitative comparison between GALA (ours, **Middle**) and GALA without anisotropic local grids (w/o $\mathcal{A}$, **Right**). The **Left** is the ground truth (GT).

## B.2 Mesh Extraction Efficiency

As in the Mosaic-SDF (Yariv et al., 2023) paper, we report the mesh extraction time at resolution $256^3$ and $512^3$ on NVIDIA A100 in Table 5. We show that with our implementation, the mesh extraction process of ours is very efficient. To further dissect the running time of our mesh extraction process, we show the flip-sign algorithm on CPU($\dagger$) consists half of the time. Further improvement can be done by adopting GPU graph traversal algorithm like (Merrill et al., 2012). The configuration is the same as stated in subsection A.4.

| Method | Resolution | Time(s) | Method | Resolution | Time(s) |
|---|---|---|---|---|---|
| 3DIGL Zhang et al. (2022) | $256^3$ | 18.44 | 3DIGL Zhang et al. (2022) | $512^3$ | 159.56 |
| S2VS Zhang et al. (2023) | $256^3$ | 5.62* | S2VS Zhang et al. (2023) | $512^3$ | 24.49* |
| Mosaic-SDF Yariv et al. (2023) | $256^3$ | 2.74 | Mosaic-SDF Yariv et al. (2023) | $512^3$ | 21.84 |
| Ours | $256^3$ | **2.39** $(1.26 + 1.23^\dagger)$ | Ours | $512^3$ | **18.73** $(8.79 + 9.94^\dagger)$ |

Table 5: Extraction efficiency comparison with Mosaic-SDF and other baselines on NVIDIA A100. $\dagger$ means the running time of the DFS-based flip-sign algorithm detailed in subsection A.1 on CPU. * means when measuring the inference time of S2VS, we conduct queries of $256^2$ or $512^2$ slice every time and loop for the whole grid in order to avoid OOM issue on GPU.

## B.3 Storage Efficiency

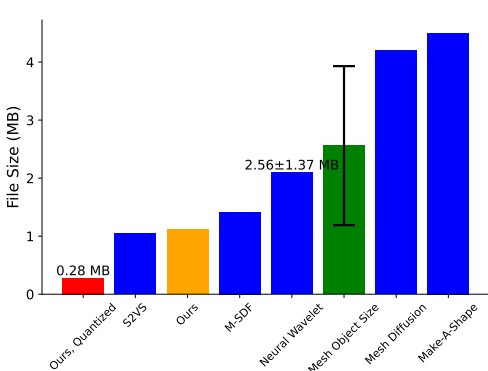

In Figure 21, we compare the file sizes of different representations in order to illusrate the high storage efficiency of our GALA. The representations to be compared in this figure, from the left to the right, are ours (GALA), S2VS (Zhang et al., 2023), M-SDF (Yariv et al., 2023), Neural Wavelet (Hui et al., 2022), raw object mesh files, Mesh Diffusion (DMTet) (Liu et al., 2023a), Make-A-Shape (Packed wavelet coefficients) (Hui et al., 2024). Raw object mesh sizes are measured on the converted water-tight meshes on ShapeNet Airplane. Each file of ours occupies only 0.28MB disk space, which is a practical benefit when processing large scale datasets.

Figure 21: Comparison of file sizes of each representation.

## B.4 Ablation Study of Overlapping Subdivision

We show the reconstruction performance by presenting Chamfer Distance (CD) of different overlap ratios $\alpha$ of subdivision in Figure 22. When $\alpha = 0$, the subdivision performed as usual octree subdivision, where a cube space is subdivided evenly into 8 cubes (here in Figure 22's 2D illustration, it is a Quadtree). To avoid making the tree too sparse at early levels while maintaining the tree structures, we expand the subdivision with ratio $\alpha$, resulting in overlapping subdivisions. However, expanding too much would decrease the granularity of each subdivision, we find a sweet spot $\alpha = 0.2$ by scanning it on the reconstruction test set.

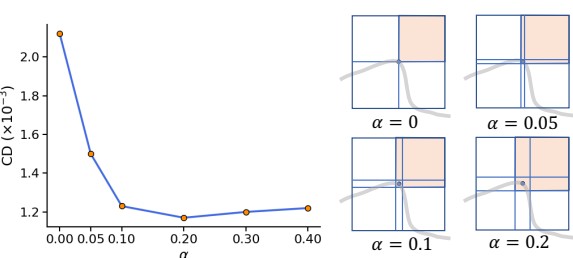

Figure 22: Ablation of child overlap $\alpha$.

# C   MORE COMPARISONS

We provide reconstruction and generation comparisons with additional baselines of XCube (Ren et al., 2024) and OctFusion (Xiong et al., 2024). XCube adopts coarse to fine sparse voxel grid generation strategy which relies on the VDB (Museth, 2013) data structure. OctFusion generates octree splits and latents encoded by VAE based on the dual octree graph network (Wang et al., 2022).

## C.1   RECONSTRUCTION COMPARISON

To show the representation ability in the sense of reconstruction accuracy, we conduct the reconstruction comparison with OctFusion (Xiong et al., 2024) and XCube (Ren et al., 2024). For fair comparison, we strictly follow the data preparation procedures of the two methods. For OctFusion, we use the provided universal autoencoder (max-depth 8) to perform the recon-

| Metrics | XCube | OctFusion | Ours(small) | Ours |
|---|---|---|---|---|
| CD $\downarrow$ ($\times 10^{-3}$) | / | 3.25 | 2.83 | 1.18 |
| LFD $\downarrow$ ($\times 10^3$) | / | 1.56 | 1.32 | 1.11 |
| #Params | 664k | 68k | 68k | 277k |

Table 6: Quantitative results of the reconstruction.

struction and for XCube, we use the separately trained autoencoders for ShapeNet airplane and chair respectively. From Figure 23 and Table 6, we can see our strong representation ability over other two baselines. Note that, since the XCube autoencoder is trained on a per-class basis, we cannot provide concrete numbers for it on our reconstruction test set, as this set includes categories that the XCube autoencoders were not trained on. In Figure 23, *No Mesh Produced* means we got an empty mesh file after running the provided script from the open-source release of XCube.

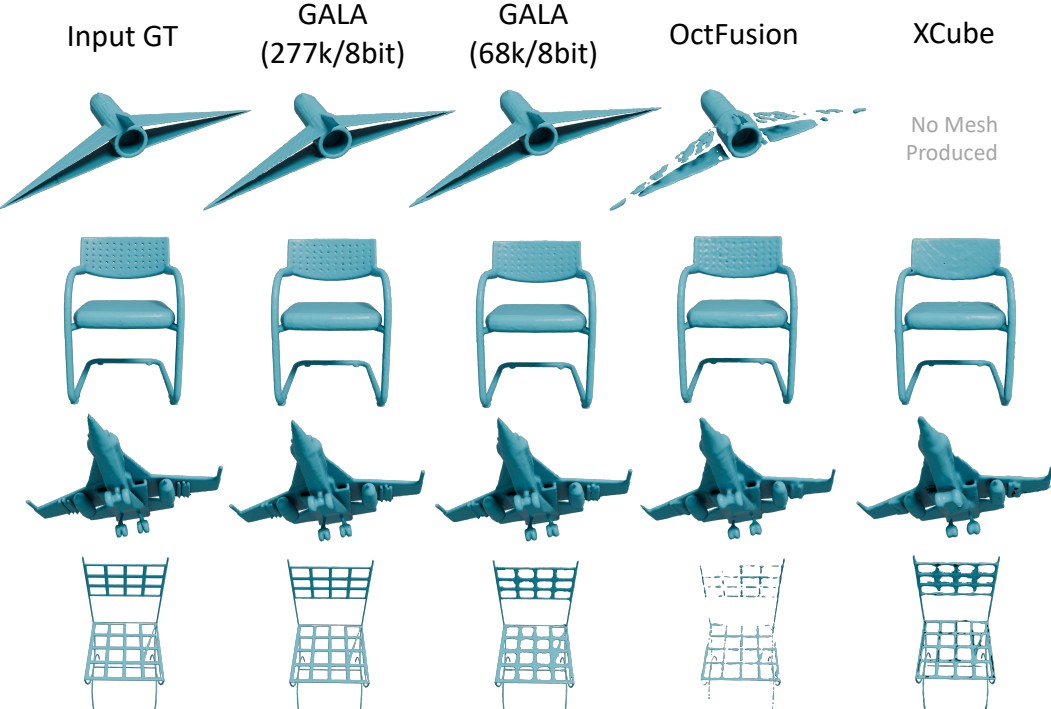

Figure 23: The qualitative comparison of the reconstruction results with OctFusion and XCube.

Regarding the numbers of parameters for OctFusion and XCube reported in Table 6, we calculate them as follows: For OctFusion (Xiong et al., 2024), $68k \approx 16^3 \times 8 + 11634 \times 3$. The $16^3 \times 8$ is the initial predicted octree split and $11634$ comes from Table 4 of OctFusion, where it reveals the average node number. For XCube (Ren et al., 2024), $664k \approx 16^3 \times 16 + 74761 \times 8$. Most hyperparameters are obtained from Table 4 of XCube's supplementary material. To account for the sparsity of the VDB structure used in XCube, we use $74761$ as the average node count, based on Table 4 of OctFusion.

## C.2 GENERATION COMPARISON

To compare the generation results, we show the qualitative comparison in Figure 24. We show more geometric details generated in the samples.

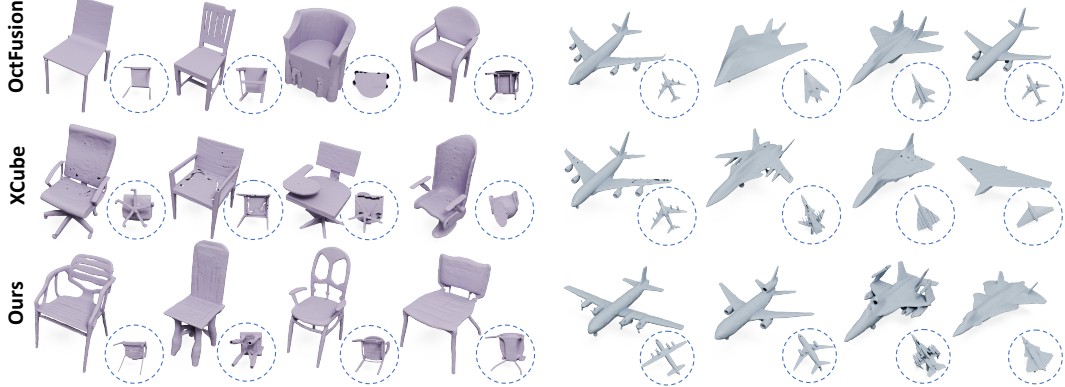

Figure 24: Qualitative comparison of generation results with OctFusion and XCube.

We do not report quantitative results here because we use different train/test splits.

## D RECONSTRUCTION ON LARGE SHAPES

To demonstrate broader extent of the representation ability of GALA, we show reconstruction results on large shapes with more than a million triangles from Stanford 3D Scanning dataset (Turk & Levoy, 1994; Curless & Levoy, 1996; Krishnamurthy & Levoy, 1996; Gardner et al., 2003). We showcase two representative object meshes from the dataset, *Lucy* (1.4M triangles, decimated from the original 28M) and *Asian Dragon* (1.6M triangles, decimated from the original 12M). As we can see from Figure 28, GALA offers detailed representation on large shapes.

## E MORE RECONSTRUCTION COMPARISONS WITH MOSAIC-SDF

Based on our own GALA implementation, we also implemented our own version of Mosaic-SDF (MSDF), where we tried to strictly follow its original setting, such as $m = 7$ for each of the axis-aligned and isotropic grid sampling. Firstly, as we can see from the left part of Figure 25, with the same low parameter budget (82k) and even the additional quantization step, our GALA represents much more faithful geometric structures such as smooth lampshade and accurate strings than MSDF, CD ($\times 10^{-3}$) 1.3 vs. 1.7. Secondly, in the right part of Figure 25, our reimplemented MSDF shows a smoother internal structures than the officially provided one but have more uneven exterior surfaces.

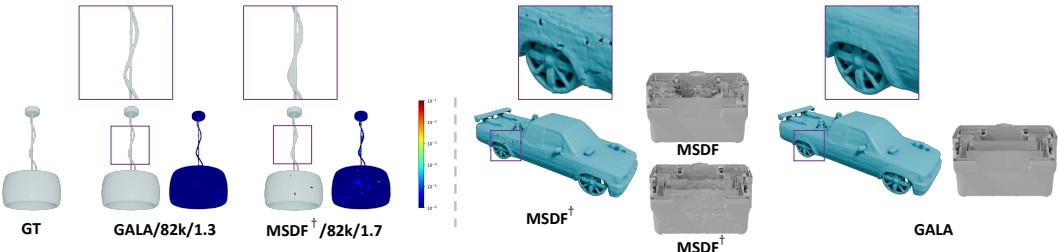

Figure 25: Compare GALA with our self-implemented Mosaic-SDF (MSDF$^{\dagger}$).

## F TEXT-TO-3D ON OBJAVERSE

To validate the ability of out GALA representation on more complicated datasets, we conduct the text-to-3D generation experiment on the Objaverse dataset (Deitke et al., 2023) and show the preliminary

results here. More specifically speaking, we use a subset of objects with matching texts (Luo et al., 2024) that contains about 130k object-text pairs due to limited time and resources. For implementation details, we use T5 (Raffel et al., 2020)-basic model as the text tokenizer and encoder as well as keep the diffusion backbone unchanged with the one we adopt in the class-conditioned generation. As we can see from Figure 26 that our GALA representations can be adapted to more complicated datasets than ShapeNet.

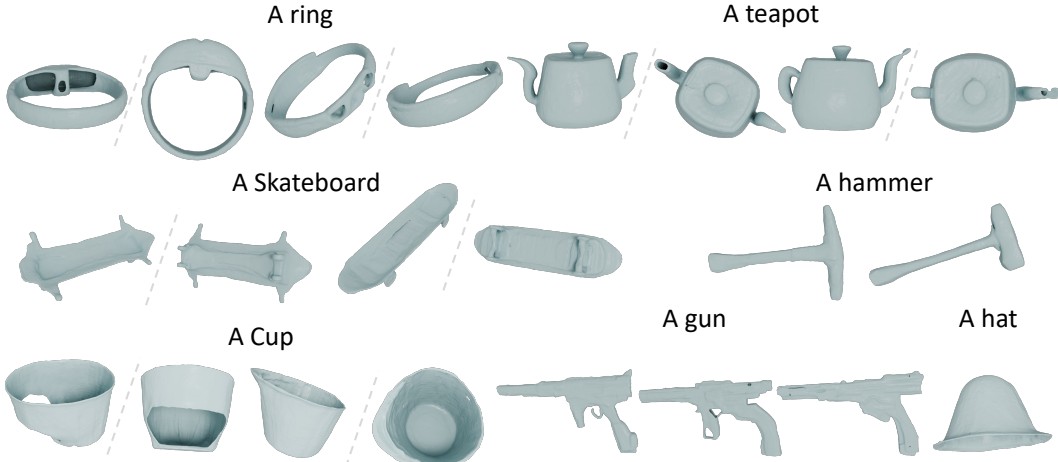

Figure 26: The text-to-3D results on a subset of Objaverse object-text pairs.

## G FAILURE CASES

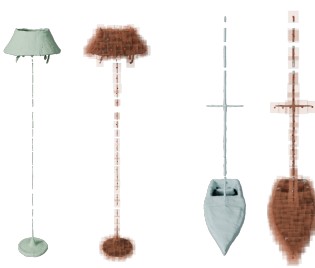

Figure 27: Examples of typical failure cases. Root voxels are visualized on the right.

At this section, we show examples of typical failure cases of the generation. As we can see from Figure 27, typical failure cases root in imperfect root voxel generation, which is the first step of our cascaded generation pipeline. Failed generation of properly *connected* root voxels would result in unconnected geometries. It will be beneficial to study how to enhance the root voxel generation, for example using more advanced diffusion training schemes. On the other hand, it is worth to note, although some of the root voxels are not properly connected, the final generated geometries are very well aligned, fine and reasonable, implying the power of GALA and our cascaded generation pipeline.

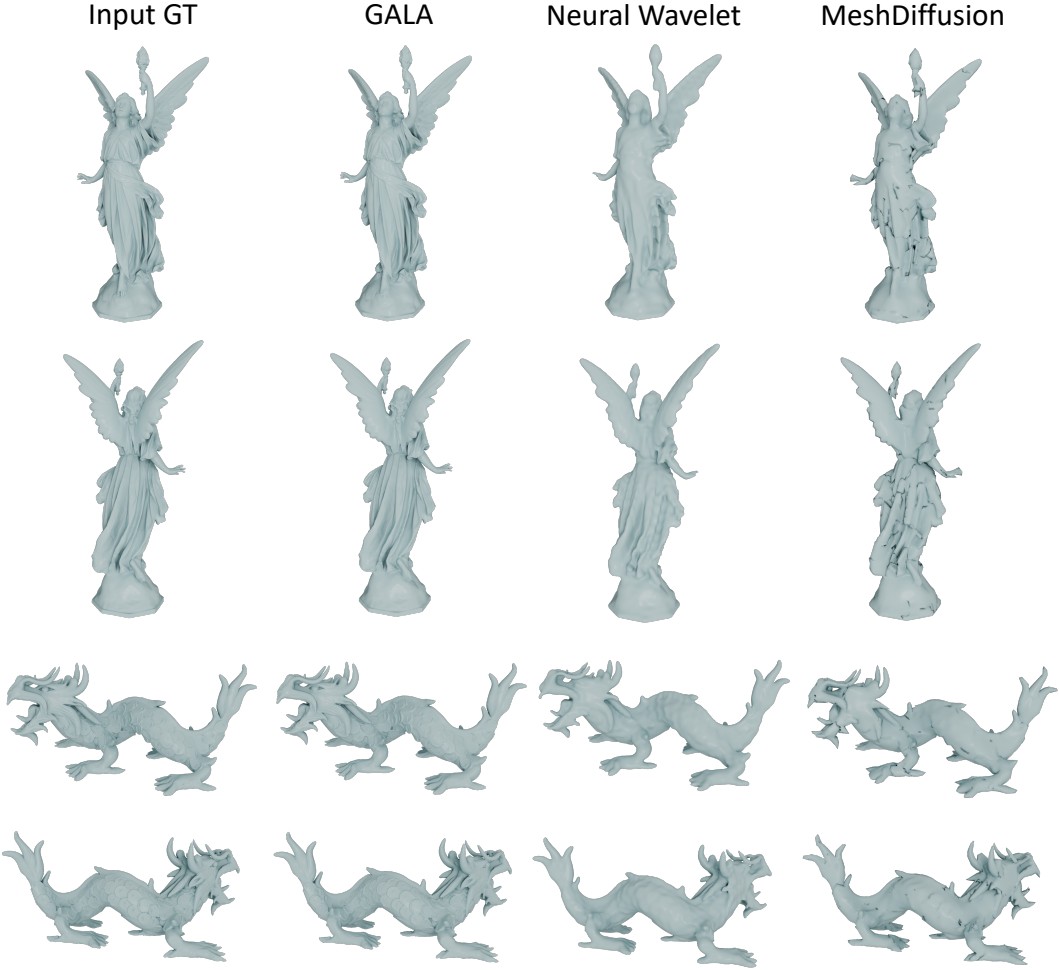

Figure 28: The comparison of reconstruction results on Stanford 3D Scanning dataset. It is apparent that GALA can reproduce details much better than the alternatives.

