# A   MORE GENERATION RESULTS

To show the full picture of our generation qualitatively, we show more uncurated generated results in Figure 2 of ShapeNet Airplane, Figure 1 of ShapeNet Chair, Figure 3 of ShapeNet Vessel and Train. As we can see from the results here, our GALA and cascaded representation are indeed capable of getting diversified and detailed generation results.

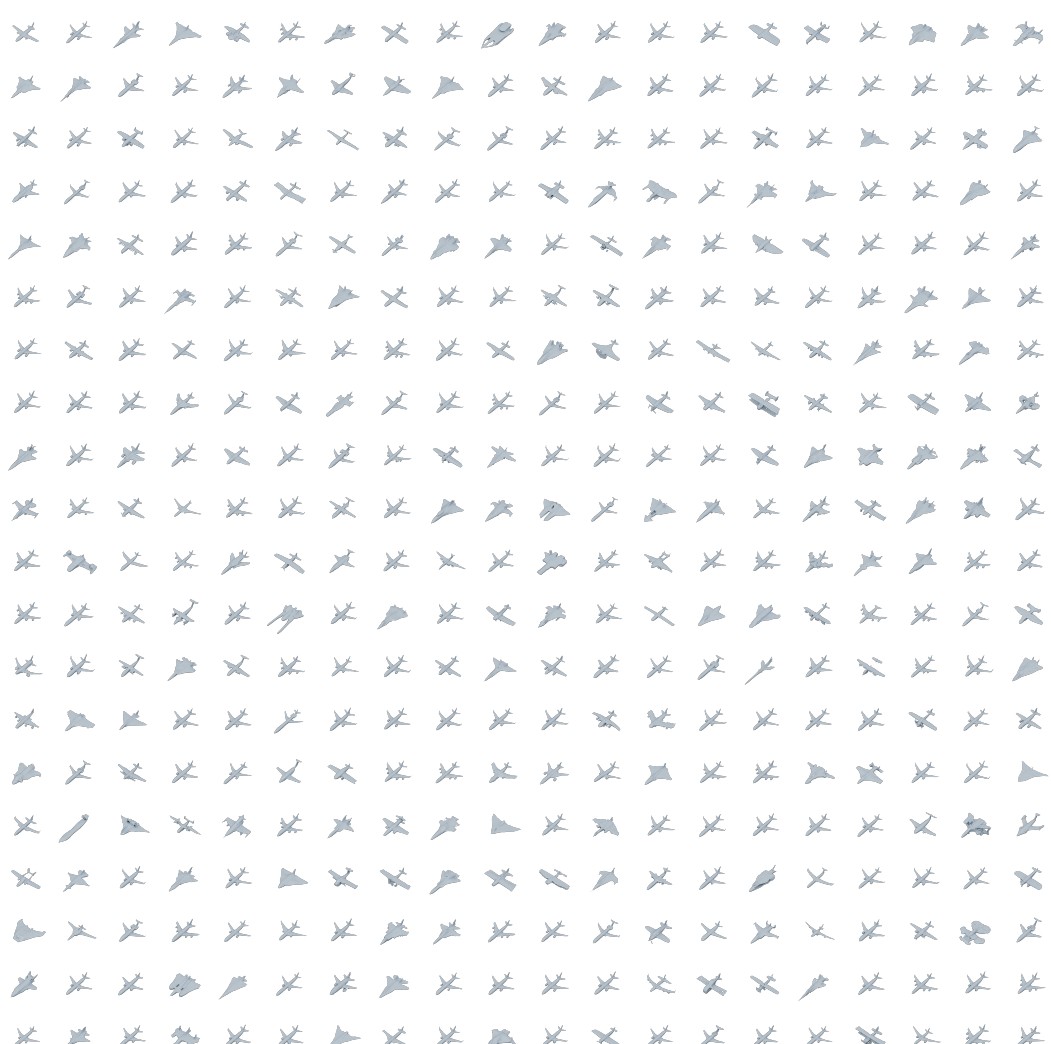

Figure 1: Uncurated 400 conditional generation results of ShapeNet Airplane.

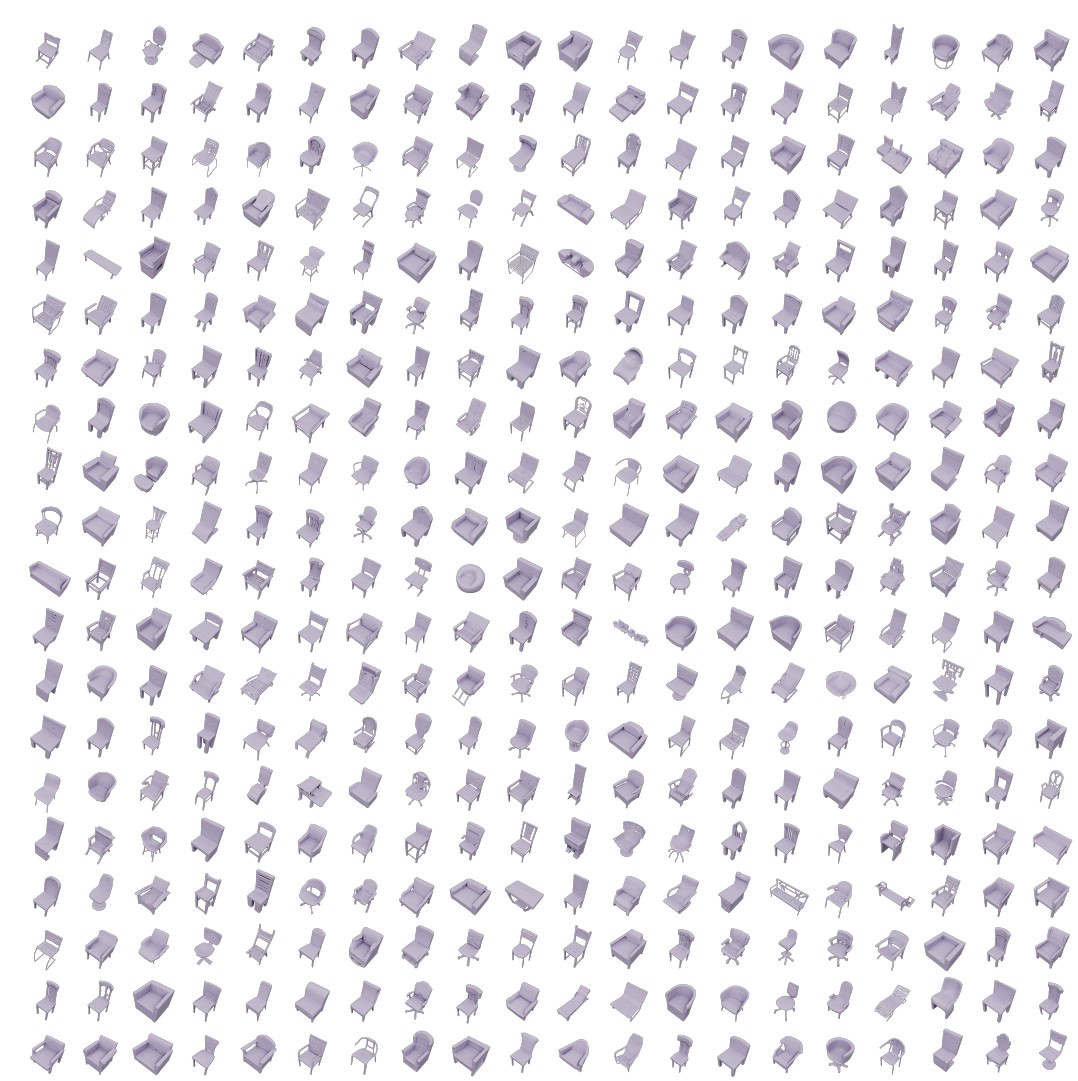

Figure 2: Uncurated 400 conditional generation results of ShapeNet Chair.

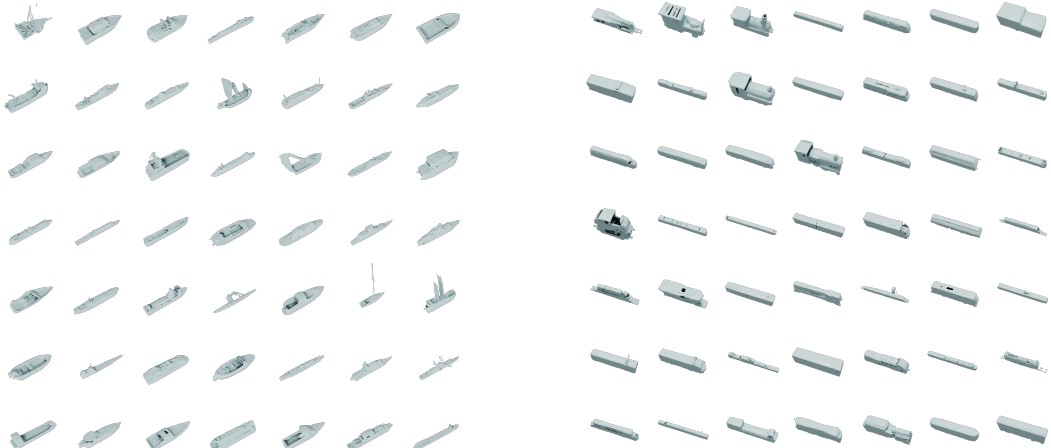

Figure 3: Uncurated 49 conditional generation results for ShapeNet Vessel and ShapeNet train.