# OpenReview forum: "GALA: Geometry-Aware Local Adaptive Grids for Detailed 3D Generation"
_ICLR.cc/2025/Conference — ICLR 2025 Poster_

### Official Review · Reviewer_bAQg · 2024-10-27

**Soundness:** 2
**Presentation:** 3
**Contribution:** 3
**Rating:** 6
**Confidence:** 4

**Summary:**

This paper describes a 3D shape representation based on trees. The input shape is initially discretized into voxels, and the location of the voxel is decided through Furthest Point Sampling (FPS). Then, each voxel is further discretized into an octree which represents the surface details. Each octree uses a local adaptive grid, with adjusted orientation and scale, to encode the local surface.
To further compress the representation, the octree and voxel information ($X_0, X_\bar{v}, X_v$) is quantized.
Finally, $X_0, X_\bar{v}, X_v$ can be generated by a diffusion model in a coarse to fine manner (first $X_0$ for the voxels, then $X_\bar{v}$).

**Strengths:**

The proposed representation is memory efficient thanks to the quantization. It cleverly exploits a combination of SDF and Octrees. Furthermore, thanks to its structure, it is possible to flatten it out and use generative models to encode it.

**Weaknesses:**

There are aspects of this method that do not convince me. I hope the authors can clarify them.
1. The main benchmark is Shape-Net, however, this dataset is known to be axis-oriented. It would be more convincing if the authors used additional datasets.
2. How well does this method scale with large shapes (e.g., shapes from the Stanford dataset)?
3. The authors claim to introduce a 3D representation, yet, they rely on a generative model to encode and decode shapes. The soundness of the representation is never evaluated. It seems that the representation is a convenient tool for model generation. If so, I would suggest the authors should reword the text.

**Questions:**

Please see the weakness section.

---

> ### Author Response · Authors · 2024-11-22
>
> 1. **More datasets other than ShapeNet are preferred:**
>
>       We have now provided additional Text-to-3D generation results on Objaverse. Please refer to Appendix F of the revised version.
>
> 2. **Large Shape Reconstruction Results:**
>
>       We have tested our GALA representation on large shapes with more than a million triangle faces from the Stanford 3D Scanning dataset. We can see from Appendix D of the revised version that our GALA-based approach can capture rich and accurate geometry details of the large and complicated shapes and outperform other baselines.
>
>
> 3. **“The authors claim to introduce a 3D representation, yet, they rely on a generative model to encode and decode shapes. The soundness of the representation is never evaluated.”:**
>
>       Thank you for your feedback. To clarify, yes, the representation is specially designed for the generation in 3D. While tightly integrated into the generative process, its soundness as a representation is evaluated through reconstruction tasks (Section 4.3) and ablation studies (Section 4.5), demonstrating its standalone capacity to model surfaces effectively. We will revise the manuscript to make this dual role clearer.

---

> > ### Author Response · Authors · 2024-12-01
> >
> > Dear reviewer, thank you once again for your feedback. We made best efforts to address your concerns, particularly evaluating our method on Objaverse and reconstructing large shapes from Stanford dataset. We hope these clarifications not only resolved the issues raised but also strengthened our paper. We would appreciate any further feedback you may provide in your evaluation of our submission.

---

### Official Review · Reviewer_s24f · 2024-11-01

**Soundness:** 4
**Presentation:** 4
**Contribution:** 3
**Rating:** 6
**Confidence:** 4

**Summary:**

The paper proposed GALA, a novel method for representing 3D shapes using local adaptive grids.  GALA's core concept is to leverage the global sparsity of surfaces within a 3D volume and their local surface properties. The representation enhances the local details by covering only the boundaries of the 3D object with a set of tree root voxels, rather than the empty space, with each voxel containing an octree to limit storage and computation to regions containing surfaces. By adjusting the orientation of the local grid and the anisotropic scales of its axes to the local surface shape, GALA can store a greater amount of detail within a given memory budget compared with Mosaic-SDF and other representations. The advantages of GALA are: (1) it effectively captures and reproduces intricate geometric and surface details, (2) it is computationally efficient, and (3) it is compatible with modern 3D generative modeling techniques, particularly those based on diffusion schemes.

Furthermore, the authors provide a cascaded generation pipeline that is capable of generating 3D shapes with significant geometric detail, demonstrating the practical application and potential of GALA in the field of 3D generation.

**Strengths:**

Originality:

1. This work's main contribution is to proposed a novel 3D shape representation which can preserve the local details by using local adaptive grids, while previous methods like Mosaic-SDF using regular grids.
2. The proposed representation is simple and efficient, achieving a good compression ratio with very little storage space and quantization operations.
3. Moreover, the authors also did comprehensive evaluation and application  like 3D shape diffusion to validate the effectiveness of the proposed representation.


Clarity:

1. This pipeline in this submission is technically sound and is largely clearly written and well organized.
2. For the comparison, the numerical results and the visual figures show a significant performance improvement over the baseline method in ShapeNet. But only a few subset with simple geometry (airplane and chair) is evaluated.
3. For the ablation study, the authors show the importance of some designs like determining orientation by bounded normals and rescaling with histogram.

**Weaknesses:**

1. While this proposed expression can enhance the details of the reconstruction, I think it is very challenging for diffusion networks to generate the explicit parameters for the root node, grid values, and grid configuration step by step. If some parameter predictions are incorrect, it can greatly affect the quality of the geometry in the subsequent steps. According to the figures in the supplementary, some structures of the generated meshes are noisy. This also limits the practical application of this expression in generative models. It would be better to discuss in the conclusion, or analyze how the errors propagate in the steps and ultimately affect the geometry's quality.

2. I think the authors have omitted some relevant citations regarding XCube and OctTree. XCube employs sparse convolution to achieve high-quality reconstruction and geometric compression, and it has also demonstrated its applicability to generative models. It would be beneficial to include additional quantitative and qualitative experiments and discussions focused on reconstruction quality, compression efficiency, and the applicability to 3D generative models.

3. The authors only tested on some simple geometric details of ShapeNet; it would be better if they could show some more complex results, like objects in Objaverse or Thingiverse.

[1] Ren, Xuanchi, et al., XCube: Large-Scale 3D Generative Modeling using Sparse Voxel Hierarchies, CVPR 2024

[2] Bojun Xiong, et al., OctFusion: Octree-based Diffusion Models for 3D Shape Generation, Arxiv 2024

[3] Objaverse: https://objaverse.allenai.org/

[4] Thingi10K: https://ten-thousand-models.appspot.com/

**Questions:**

1. I noticed that the paper lacks a comparison with the reconstruction results of Mosaic-SDF, only comparing the generation results with it in Table 3. After reading the paper of Mosaic-SDF and I found that they also lack this experiment. However, my understanding is that "Ours w/o A" in Table 2 should be a structure similar to Mosaic-SDF. Are there any other differences?


2. In Figure 7, when comparing the results with Mosaic-SDF, I didn't notice much difference. Can you provide more representative examples in the revised version? Or is this the maximum performance gap compared with Mosaic-SDF?

---

> ### Author Response · Authors · 2024-11-22
>
> 1. **Error propagation through steps of cascaded generation:**
>
>     Good question. Our method, along with many other sequential or cascaded generations, including the two you mentioned (XCube and OctFusion), all need to address error accumulation due to the training and inference disparity. We mainly rely on noise augmentation used in previous works [1, 2, 3], where noise is injected to GT conditions by the diffusion forward process with random noise levels. From our results as well as others, it seems that the error can be effectively managed using similar techniques.
>
> 2. **Comparison to OctFusion and XCube:**
>
>      We now show reconstruction and generation comparisons with OctFusion and XCube in Appendix C of the revised version, demonstrating that our method leads to more accurate and efficient representation than the two baselines, as well as its capability of generating better geometric details. Detailed evaluation protocols can be found in Appendix C.
>
>
> 3. **Training on more complicated datasets such as Objaverse dataset:**
>
>     We are now showing additional text-to-3D experiments in Appendix F of the revised version. This demonstrates our ability to adapt to larger and more complicated datasets.
>
> 4. **What is the difference between Mosaic-SDF and “Ours w/o A”?**:
>
>     Yes, “Ours w/o A” is similar to Mosaic-SDF except that we still have the additional *octree forest* setting in “Ours w/o A”.
>
>
> 5. **More comparisons with Mosaic-SDF:**
>
>     In Figure 7 of the main paper, it is noted that we can model complicated geometries such as the interior of the car model with fewer and quantized parameters than Mosaic-SDF. We also provide additional Mosaic-SDF comparisons in Appendix E of the revised version. In Appendix E, we can see that GALA can represent complex geometry more accurately both in low and default parameter counts.
>
> [1] Ho, Jonathan, et al. "Cascaded diffusion models for high fidelity image generation." Journal of Machine Learning Research 23.47 (2022): 1-33.
>
> [2] Saharia, Chitwan, et al. "Photorealistic text-to-image diffusion models with deep language understanding." Advances in neural information processing systems 35 (2022): 36479-36494.
>
> [3] Xu, Xiang, et al. "Brepgen: A b-rep generative diffusion model with structured latent geometry." ACM Transactions on Graphics (TOG) 43.4 (2024): 1-14.

---

> > ### Comment · Reviewer_s24f · 2024-11-29
> >
> > Thank you for your clarifications. I was positive to this work and remain the positive score.

---

### Official Review · Reviewer_fRPo · 2024-11-03

**Soundness:** 4
**Presentation:** 2
**Contribution:** 3
**Rating:** 8
**Confidence:** 2

**Summary:**

The authors propose a new geometric representation based on storing optimized SDF values of a shape on an adaptive, overlapping grid of optimally scaled and rotated cells. This representation can be very efficiently generated for any shape and enables diffusion-based generation tasks.

**Strengths:**

I congratulate the authors for their submission. They build on existing work to propose a non-trivial, novel geometric representation that is particularly useful for detailed 3D generative tasks. While I am not an expert in 3D generative modeling, it seems clear that this work outperforms the state of the art in resolutive power, and outperforms its main competitor (Mosaic-SDF) on reconstruction tasks. There is little doubt in my mind that this work would be received positively in the neural geometric processing community.

**Weaknesses:**

The main weaknesses of this work are in the magnitude of the contribution and the clarity of the writing:

Contribution. It could be argued this work is limited to being an adaptive version of Mosaic-SDF, and that most choices made in the algorithm are the same choices any researcher would made if attempting to extend M-SDF to an adaptive realm (the authors of M-SDF themselves even mention “adding local coordinate frames” as a potential future work avenue). The authors of this work address this and also mention the cascading generation strategy as a difference with M-SDF; however, the magnitude of this contribution is not particularly explored: what would have happened if they used an identical flow-matching strategy to the one in M-SDF?

Clarity of writing. Several key aspects of the work are missing or unclear, to the point in which I don’t believe I understand important sections of the algorithm (see “Questions” below). Additionally, perhaps due to LaTeX template changes, groups of citations that should be grouped in parentheses are not, making reading difficult (e.g., 053, 080). The code link also seems to be missing.

Nonetheless, I do not think these weaknesses justify rejecting this work, which undoubtedly outperforms the state of the art and will definitely save authors’ time if they wish to improve the M-SDF performance in the future.

**Questions:**

- How is each grid subdivided? As far as I can tell, the only explanation of this subdivision is “we recursively deduce the further levels of tree nodes in an overlapping way where we slightly expand the divided spaces at each level by ratio α.” Which is far from enough detail for a reader to replicate the results. Which cells are subdivided? According to which logic? Along which axis?
- How is a mesh extracted from this adaptive grid? Are the authors setting up some regular grid and interpolating values at grid cells using Eq. (2), and then using Marching Cubes or some other reconstruction algorithm? Or are they somehow extracting a mesh directly from the stored SDF data at the adaptive cells? The latter seems very non-trivial and would be an important contribution on its own.
- What does “the weighting function which is defined by the infinity ball centered at the grid center pig and rotation matrix Oi” mean? Isn’t \omega(x,i) just Eq. (3)?
- It was very hard for me to follow Sec. 3.2., I think partly because of the notation used: Are x_v (lowercase) and X_V (uppercase) the same thing? Am I right in understanding that in this cascading scheme, given an initial tree X_0, the grid configuration X_V and the SDF values at those points will not be affected by the conditioning? Since the class label is only involved in Step 1.

---

> ### Author Response · Authors · 2024-11-22
>
> 1. **What if we use the same Flow Matching strategy as Mosaic-SDF**:
>
>      Flow Matching (FM)  [1] has shown some advantages over general diffusion schemes. Both in theory and practically, FM should be adaptable to our generation training process. The reasons we chose v-prediction diffusion as our generation training scheme are: (1) At the early development stage, diffusion training frameworks were more mature and easy to get access to; (2) Other baselines other than Mosaic-SDF adopted diffusion, and Mosaic-SDF itself did not conduct an ablation study to show performance boost by shifting usual diffusion paradigm to FM in its specific case. Adopting FM to GALA would be interesting to explore in future work.
>
> 2. **The grouping of citations:**
>
>       We are sorry about this. It is fixed now.
>
> 3. **How is each grid subdivided**: In Figure 1, as well as in the introduction, the subdivisions are based on octrees, where the space in a cube is evenly subdivided into 8 smaller cubes. After recursive subdivision to a certain depth, adaptive grids will be extracted in non-empty nodes (green ones visualized in Figures 1 and 3). We have made this information more consistent and clear by changing the wording in Sec 3.1.1 and Figure 3 as well as adding more explanations in Appendix B.4, in the revised version.
> How is a mesh extracted from this adaptive grid: We did the former way as you mentioned, by first extracting values within locations of a certain resolution regular grids by interpolation via Equation 2. We have addressed this concern in Sec 3.1.3 as well as Appendix A.1, in the revised version.
>
> 4. **What does “the weighting function which … rotation matrix $O_i$” mean**:
>
>       Sorry for any confusion. Yes, it is about Equation 3. What we want to explain here is that the neighborhood, on which the weight function is defined, is determined by $\ell_\infty$-norm centered at center $\mathbf{p}_g^i$, orientated by $\mathbf{O}_i$ and scaled via $\mathbf{s}_i$. We have changed the wording there.
>
> 5. **Is $\mathbf{x}_{V}$ and $X_V$ the same thing?**:
>
>       $X_V$ means a set of vectors and $x_{V}$ means a single vector, and $x_{V} \in X_V$. To clarify this further, every node’s information will be flattened into a vector. Since we have multiple nodes, each level of information forms a set of vectors. We have made this part more clear in the revised version, please refer to line 302-305 of Sec 3.2.
>
>
> 6. **The later two stages will not be affected by the conditioning?**:
>
>      No. All three stages will be conditioned on the input labels. As formulated via $P(X_o|y)$, $P(X_{\bar{V}}|X_o,y)$ and $X_V = F(X_{\bar{V}},y)$ in lines of 320, 322 and 323, there are $y$ labels in all three equations. You can also find more detailed figure illustration in Figure 19 of Appendix A.5.
> [1] Lipman, Yaron, et al. "Flow matching for generative modeling." arXiv preprint arXiv:2210.02747 (2022).

---

> > ### Comment · Reviewer_fRPo · 2024-11-25
> >
> > Thank you very much for your clarifications! I was already very positive about this work, and I remain equally positive.

---

> > > ### Author Response · Authors · 2024-11-28
> > >
> > > Thank you for your positive remarks and encouraging feedback!

---

### Official Review · Reviewer_3cms · 2024-11-04

**Soundness:** 3
**Presentation:** 4
**Contribution:** 3
**Rating:** 8
**Confidence:** 5

**Summary:**

The paper presents a storage-efficient new representation of 3D triangle surface meshes, GALA, which can be used for 3D generation trained by a transformer. To create the proposed representation, voxel tree root nodes are initialized over the surface of the mesh using FPS. In the local adaptive grid extraction step, the grid orientation is aligned with the geometric structure using PCA on the local normal vectors, and the grid is rescaled using a histogram. With this representation, the authors show that 3D reconstruction and generation can be performed using cascaded diffusion models. Experiments demonstrate the effectiveness of the proposed method in 3D reconstruction and generation, both quantitatively and qualitatively.

**Strengths:**

(1) The proposed representation has a low number of parameters, fast fitting time, high precision, and quick mesh extraction time, implemented using pure CUDA and LibTorch.

(2) Initially, I think that the proposed representation is sensitive to the FPS results regarding the number of root tree nodes ($N_o$) and overlap ratios ($\alpha$), so I doubt whether it can effectively cover the thin parts of meshes. However, the authors conduct numerous ablation studies on design choices to achieve optimal performance for general meshes.

(3) The results show that the proposed method achieves SOTA performance on reconstruction and generation with a few number of parameters compared to existing baselines.

(4)  The paper has high writing quality which makes it easy to understand.

**Weaknesses:**

(1) As shown in the Sec. C. Failure Cases, the proposed representation cannot cover the long thin mesh parts.

**Questions:**

I found that the authors have currently created an anonymous repository, and I am curious whether the authors intend to make the code public in the future.

---

> ### Author Response · Authors · 2024-11-22
>
> 1. **Failure cases show the method cannot cover long thin parts**:
>
>      The inability to fully handle very thin features is common to all methods, e.g., see the last row of Fig. 6 for some reconstruction results. Whatever the method, its representational capacity will always be limited while there will always be features that are too small to capture. Fig. 6 also shows a typical example, that our method with GALA outperforms the competitors in this regard.
>
>       Note that the imperfect result that the reviewer was looking at, in Fig. 27, was that of generation. As explained in Appendix G, the disconnections in the thin part during generation were caused by the disconnected root voxels that were generated, while the thin structures were correctly captured within each root voxel. More advanced generation techniques, which are parallel to our current contribution, may help us cover long thin parts better in the generation phase. We leave that for future work.
>
> 2. **Will the code be released publicly in the future?**: Yes, the code and any data we used will be made publicly available upon paper acceptance.

---

> > ### Comment · Reviewer_3cms · 2024-11-25
> > **Response to rebuttal**
> >
> > Thank you for your response. I have read it, as well as other reviews.
> >
> > I understand that handling long, thin parts is extremely challenging, so I don’t consider this a major issue. I appreciate the additional experiments conducted for the rebuttal, especially since Mosaic-SDF hasn’t released their official code and the author implemented their own version. Therefore, I will maintain my original rating of 8.

---

> > > ### Author Response · Authors · 2024-11-28
> > >
> > > Thanks for your thoughtful review and positive response!

---

### Author Response · Authors · 2024-11-22
**General Response for Rebuttal**

We thank all reviewers for their generally positive and detailed feedback, as well as the constructive suggestions. We are pleased with the recognized strengths of our paper, including the novel GALA representation, its efficiency and effectiveness in 3D generation and reconstruction, and the strong quantitative and qualitative results. Reviewers also appreciated the overall quality and clarity of the writing, as well as the comprehensive ablation studies and comparisons. Still, we would take this opportunity to address some common concerns here and then individual comments in reviewer-specific rebuttals.

**Common Concerns and responses:**
1. **Comparison to baselines: Reviewer s24f asked to provide additional comparisons to other baselines (XCube, OctFusion)**.

      We have now added comparisons to OctFusion and XCube, demonstrating GALA’s advantage of accurate and efficient reconstruction and generating detailed geometries over these newly suggested baselines. Additionally, we are providing more reconstruction comparisons with Mosaic-SDF, further clarifying the differences. See Appendix C and E of the revised version for details

2. **More datasets: Reviewers s24f and bAQg suggested training on datasets other than ShapeNet to further demonstrate the representational power of GALA.**

    We conducted experiments on more complex datasets, such as Objaverse and the Stanford 3D Scanning dataset, showing GALA’s ability to handle larger and more varied, as well as more detailed, geometries. See Appendix F and D of the revised version.


3. **Further clarity: Specific parts of the paper, such as grid subdivision logic, mesh extraction methodology, and cascading diffusion, were noted as being a bit unclear (fRPo, bAQg).**

      Key sections, including the grid subdivision logic, mesh extraction methodology, and the cascading generation process, were revised to improve clarity. These changes are reflected in Sec. 3.1.1, 3.1.3, and 3.2, as well as Appendix A.1 and B.4.
Detailed responses to individual reviewers appear below. Again, we sincerely thank the reviewers for their thoughtful comments, which have helped us to further improve the quality and clarity of our submission.

---

### Meta-Review · Area_Chair_dHhP · 2024-12-17

**Metareview:**

In this paper, the authors propose a novel representation of 3D shapes called GALA, which is able to represent complex geometric details with efficient memory cost. The method leverages the global sparsity of surfaces within a 3D volume and their local surface properties, and builds a series of bounding boxes with local anisotropic SDFs. The paper provides both qualitative and quantitative results and compares them with several existing baseline methods. The paper is well-written and clearly presented.

However, the contribution of the paper is incremental, as each individual component is widely used in existing works. In addition, as mentioned by the authors, there are issues in representing certain shapes, such as long and thin structured shapes. Therefore, I recommend acceptance of the paper, but as a spotlight paper.

**Additional Comments On Reviewer Discussion:**

The reviewers initially raised concerns about the insufficient comparisons to existing methods, the lack of certain technical details, the generalization of the training data, and the limitations of the work. The authors have successfully addressed the first three issues to the reviewers' satisfaction. However, the concern regarding the limitations of the work has not been fully addressed and remains an area for improvement.

---

### Decision · Program_Chairs · 2025-01-22

Accept (Poster)